# Condition Number Based Low-Bit Quantization for Image Super-Resolution

Kai Liu [* 1]  Dehui Wang [* 1]  Zhiteng Li [1]  Zheng Chen [1]  Yong Guo [2]  Linghe Kong [† 1]

## Abstract

Low-bit model quantization for image super-resolution (SR) is a longstanding task that is renowned for its surprising compression and acceleration ability. However, accuracy degradation is inevitable when compressing the full-precision (FP) model to ultra-low bit widths ($2 \sim 4$ bits). Experimentally, we observe that the degradation of quantization is mainly attributed to the quantization of activation instead of model weights. Considering that the activation quantization error is hard to minimize, minimizing the impact of the error emerges as a good choice, which is described by the condition number. Therefore, we propose **CondiQuant**, a condition number-based low-bit post-training quantization (PTQ) for image super-resolution. Specifically, we formulate the quantization error of activation as the condition number of weight metrics. By decoupling the representation ability and the quantization sensitivity, we design an efficient proximal gradient descent algorithm to iteratively minimize the condition number and maintain the output. With comprehensive experiments, we demonstrate that CondiQuant outperforms existing PTQ methods in accuracy without computation overhead and gains the theoretically optimal compression ratio in model parameters. Our code is available at https://github.com/Kai-Liu001/CondiQuant.

## 1. Introduction

Image super-resolution (SR) aims to restore the high-resolution (HR) images from the low-resolution counterparts. It is a foundational computer vision task in low-level vision and image processing, widely studied in medical imaging (Greenspan, 2008; Isaac & Kulkarni, 2015; Huang et al., 2017), surveillance (Zhang et al., 2010; Rasti et al., 2016), remote sensing (Bandara & Patel, 2022), and mobile phone photography (Wu et al., 2024). Nonetheless, the

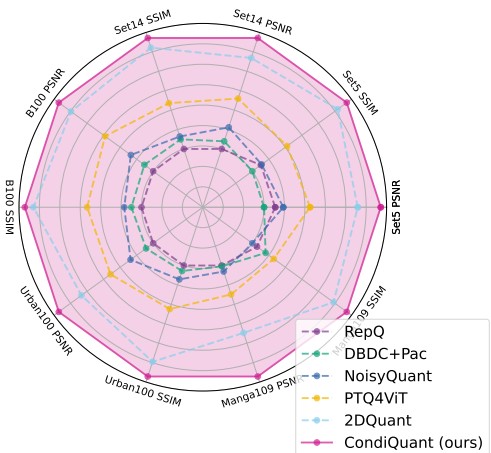

*Figure 1.* Comparison with SOTA PTQ methods on five benchmarks. Our CondiQuant gains consistently better performance.

existing edge devices' notorious limited computation and memory ability hinder real-world deployment. Therefore, it is increasingly urgent to develop model compression and acceleration techniques for SR models to reduce the redundancy in both model parameters and inference computation.

Model quantization (Choukroun et al., 2019; Ding et al., 2022; Hubara et al., 2021; Li et al., 2021) is a powerful compression technique that compresses the model from full-precision to low-bit representations. With quantization, the time-consuming floating-point operations are converted into efficient integer ones, making it an ideal candidate for model compression in resource-constrained edge devices. However, the conversion inevitably leads to severe performance degradation, especially when compressing to ultra-low bit width ($2 \sim 4$ bits). The situation is more severe in vision transformers (ViTs) due to the deterioration of self-attention.

The two branches of quantization, aka quantization-aware training (QAT) and post-training quantization (PTQ), deal with the degradation in different ways. QAT, a firm adherent of backpropagation, concurrently optimizes the model weights and quantizers' parameters via straight-through estimator (STE) (Courbariaux et al., 2016). Though inspiring results are obtained, the ineluctable demand for large datasets and long-time training discourages the deployment. Whereas PTQ is much more efficient. Most PTQ methods utilize efficient algorithms to calculate the quantizers' parameters in a small calibration set. Given the complexity of both branches, PTQ is usually inferior to QAT and calls for

[1]Shanghai Jiao Tong University, Shanghai, China [2]Huawei. Correspondence to: Linghe Kong <linghe.kong@sjtu.edu.cn>.

*Proceedings of the 43rd International Conference on Machine Learning*, Seoul, South Korea. PMLR 306, 2026. Copyright 2026 by the author(s).

advanced solutions. To improve the inferior performance of PTQ, we observe two key experimental phenomena. The first observation is about the attribution of degradation.

> **Observation 1:** *The degradation of quantization is mainly attributed to* **activations**.

Therefore, we leverage condition number, a metric in numerical analysis, to reflect the changes caused by the quantization of activations. In detail, condition number could measure how much the output value can change given a disturbance in the input value, *i.e.,* the quantization error of activation. Considering a small disturbance, the smaller the condition number is, the smaller the output varies. Thereafter, one way to reduce the quantization error is by minimizing the condition number of the weight matrix. Inspired by MagR (Zhang et al., 2024), we verify that their observation in LLMs also holds in SR models, which is

> **Observation 2:** *The feature matrix $X$ across all linear layers is approximately* **rank deficient**.

In a linear layer, the output $Y$ can be expressed as matrix multiplication $Y = XW$, where $W$ is the weight matrix and $X$ is the feature matrix. When $X$ is rank-deficient, there are infinite solutions for $W$ that satisfy the equation. This desirable property allows a shift in the weight matrix to quantization friendliness while maintaining the output.

Based on the previous two observations, we focus on the impact of quantization of activation, establish its relationship with condition number, formulate the optimization problem, and derive proximal gradient descent to solve it efficiently. With the proposed CondiQuant, the restoration accuracy is improved evidently as shown in Fig. 1.

To sum up, our main contributions are fourfold:

1. We design CondiQuant, a novel PTQ method for image SR based on condition number. CondiQuant gains theoretically optimal compression and speedup ratios with the currently minimum performance degradation.

2. We propose that the model's sensitivity to quantization is related to the condition number of weights in the linear layers and could be optimized efficiently.

3. We design an efficient algorithm based on proximal gradient descent to reduce the condition number of the weights while approximately maintaining the output.

4. Comprehensive comparison experiments are conducted to show the SOTA performance on efficiency and effectiveness of our proposed CondiQuant. Besides, extensive ablation studies are conducted to prove the robustness and efficacy of our proposed CondiQuant.

## 2. Related Work

**Image Super-Resolution.** The trailblazing research on image super-resolution with deep neural network is SR-CNN (Dong et al., 2016), which utilizes convolution layers to replace conventional methods in SR. Since then, dazzling network architectures (Chen et al., 2022; 2023; Lim et al., 2017; Zhang et al., 2018a) are designed to achieve more stirring image restoration accuracy. RDN (Zhang et al., 2018b) designs dense skip connection and global residual to utilize abundant local and global features. With the advance of vision transformer, SwinIR (Liang et al., 2021) uses window-attention to fully make use of global information. As reconstruction accuracy increases, researchers focus more on model parameters and operations. Generally, the ViT-based SR networks are much smaller and faster than pure CNNs. However, even the advanced SR networks like SwinIR, current SR models are still too large to be deployed on resource-constrained edge devices. Therefore, research of quantization on ViT-based networks is in urgent need to make SR possible on mobile devices.

**Post-Training Quantization.** PTQ is famous for its fast speed and low cost during quantization. Recently, excellent PTQ methods have been proposed to advance the performance and efficiency of PTQ. As a PTQ method specifically for ViT, PTQ4ViT (Yuan et al., 2022) proposed twin uniform quantization to reduce quantization error on the output of softmax and GELU. RepQ (Li et al., 2023) decouples the quantization and inference processes and ensures both accurate quantization and efficient inference. NoisyQuant (Liu et al., 2023) surprisingly finds that adding a fixed Uniform noisy bias to the values being quantized can significantly reduce the quantization error. However, most of the above PTQ methods are only for Transformer blocks and the generalization ability on SR tasks is relatively low. Whereas, 2DQuant (Liu et al., 2024) searches the clipping bounds in different ways on different distributions and is currently the best PTQ method on the SR task. However, most of the above methods only concentrate on the quantization process and ignore adjustments in the weights.

**Condition Number.** The concept of the condition number originated and developed in numerical analysis (Turing, 1948; von Neumann & Goldstine, 1947). It measures how much the output value of the system or function can change for a small change in the input. A matrix with a high condition number is said to be ill-conditioned and leads to huge output changes given a small disturbance in input. It also plays an important role in machine learning. Freund proposed optimization algorithms generally converge faster when the condition number is low (Freund et al., 2018). Additionally, the condition number is leveraged to enhance the numerical stability of regression models by fine-tuning hidden layer parameters (Xiao et al., 2018). In this work, we formulate the optimization of quantization error via condition number and propose CondiQuant to solve it efficiently.

# 3. Method

## 3.1. Preliminaries

Given an SR network, we use upper and lower clipping bounds to quantize weights and activations (Liu et al., 2024). Usually, we leverage the fake quantization to simulate the quantization process with no simulation error. The fake-quantize process can be written as:

$$x_c = \text{Clip}(x, l, u), x_r = \text{R}(s(x_c - l)), x_q = sx_r + l, \quad (1)$$

where $x$ denotes weight $w$ or activation $a$ being quantized and $s = (2^N - 1)/(u - l)$ is the quantization ratio. $l$ and $u$ denote the lower and upper bounds for clipping, respectively. $\text{Clip}(x, l, u) = \min(\max(x, l), u)$ constrains the input to be between $l$ and $u$, and $\text{R}(\cdot)$ rounds the input to nearest integer. With Eq (1), $x$ is dispersed into discrete candidates.

## 3.2. Analysis

To analyze the loss attribution in quantization, we rewrite the linear layer with quantization as follows:

$$\begin{aligned}\hat{Y} := \hat{X}\hat{W} &= (X + \delta X)(W + \delta W) \\ &= XW + X\delta W + \delta XW + \delta X\delta W,\end{aligned} \quad (2)$$

where $\hat{X}$ denotes the quantized value and $\delta X$ denotes the quantization error of $X$. The second-order term $\delta X\delta W$ can be ignored due to its tiny impact on performance. Thus Eq. 2 can be rewritten as:

$$\hat{Y} \approx XW + X\delta W + \delta XW = Y + X\delta W + \delta XW. \quad (3)$$

Shown in Tab. 1, we observe that the influence of $X\delta W$ and $\delta X\delta W$ is smaller than $\delta XW$.

This result can be explained by two reasons. **First**, the activation varies with different inputs while the weight is fixed. Therefore, the average value of the absolute value of the $\delta X$ elements is much greater than that of $\delta W$. **Second**, the $W$ is sensitive to minor changes, *i.e.*, $\delta X$. Hence, we only concentrate on $\delta XW$ and ignore $X\delta W$ in the following discussion. So we can further approximate the loss:

$$\hat{Y} \approx XW + \delta XW, \quad \delta Y \approx \delta XW. \quad (4)$$

4 indicates that the degradation of model performance can be divided into two components: (1) the magnitude of activation quantization error and the sensitivity of the weight to $\delta$. To minimize the approximate loss, a naive solution is to reduce the Frobenius norm of $\delta X$ *i.e.*, $||\delta X||_F$. However, it is impractical as $\delta X$ is input-related, and rounding-off errors are impossible to reduce, especially with low bits. Therefore, we focus on the model's sensitivity to quantization, related to condition number.

## 3.3. CondiQuant

We begin with the observation of rank deficiency of the feature matrix. Then we derive the optimization objective and illustrate the methodology to arrive at the solution.

**Rank Deficiency.** We visualize the rank ratio in Eq. 2. Most of the layers are rank-deficient, and only two layers are full rank. The mean value of the rank ratio is 0.626. In the linear

| | $Y$ | $Y + X\delta W$ | $Y + \delta XW$ | $Y + \delta X\delta W$ |
|---|---|---|---|---|
| PSNR | 32.2543 | 32.0117 | 31.3304 | 32.2519 |
| SSIM | 0.9293 | 0.9270 | 0.9221 | 0.9293 |

*Table 1.* Attribution of quantization loss. $Y$ denotes the FP model and the rest denotes performing quantization on $W$, $X$, and both.

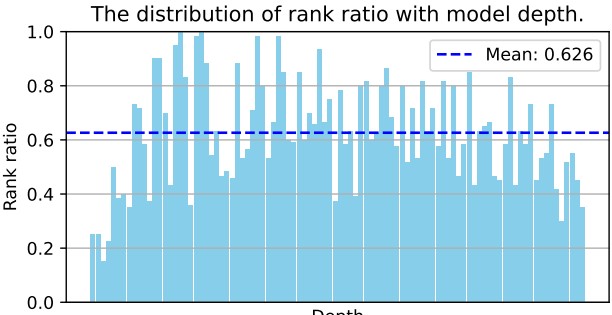

*Figure 2.* Distribution of activation ranks along the ($\times 2$) model depth. Most activations are severely rank-deficient.

layer, the process is matrix multiplication, *i.e.*, $Y = XW$. If $W$ is rank deficient, there exist infinite $W'$ that satisfies $XW = XW'$. Hence, this result serves as a theoretical foundation to select $W'$ to minimize $\delta XW$.

**Formulation.** In this section, we formulate the relationship between model sensitivity and condition number. We begin with $\|\delta Y\|_2$ and $\|Y\|_2$:

$$\|\delta Y\|_2 = \|\delta XW\|_2 \leq \|\delta X\|_2 \|W\|_2, \quad (5)$$

$$\|Y\|_2 = \|XW\|_2 \geq \|X\|_2 \, \sigma_{min}(W), \quad (6)$$

where $\|\cdot\|_2$ is the bi-norm of the matrix, and $\sigma_{min}(\cdot)$ denotes the minimum singular value. With Eq. 5, we can establish the relationship between quantization loss and condition number:

$$\frac{\|\delta Y\|_2}{\|Y\|_2} \leq \frac{\|\delta X\|_2 \|W\|_2}{\|X\|_2 \, \sigma_{min}(W)} = \kappa(W)\frac{\|\delta X\|_2}{\|X\|_2}, \quad (7)$$

where $\kappa(W)$ denotes the condition number of $W$. This formula shows that under the same rounding-off error, the impact of quantization is greater with larger $\kappa(W)$. Detailed derivation can be found in the supplementary materials. Therefore, we minimize the condition number of $W$ and maintain the output. The objective can be written as:

$$\min_W \kappa(W) = \frac{\sigma_{\max}(W)}{\sigma_{\min}(W)}, \text{s.t.}||XW - X\hat{W}||_F \leq \epsilon, \quad (8)$$

where $\epsilon$ is a small positive number that limits the magnitude of the loss. However, it is a non-convex and non-smooth optimization when minimizing the condition number directly. To address this, we construct a proxy objective to optimize and propose the proximal gradient method to solve it.

**Proxy objective function.** To minimize the condition number, our strategy is to compact the distribution of all singular values more densely. Specifically, we design a regularization term to minimize the deviation of singular values:

$$\min_W \frac{1}{2}\|XW - Y\|_F^2 + \lambda \cdot \mathcal{G}(W), \quad (9)$$

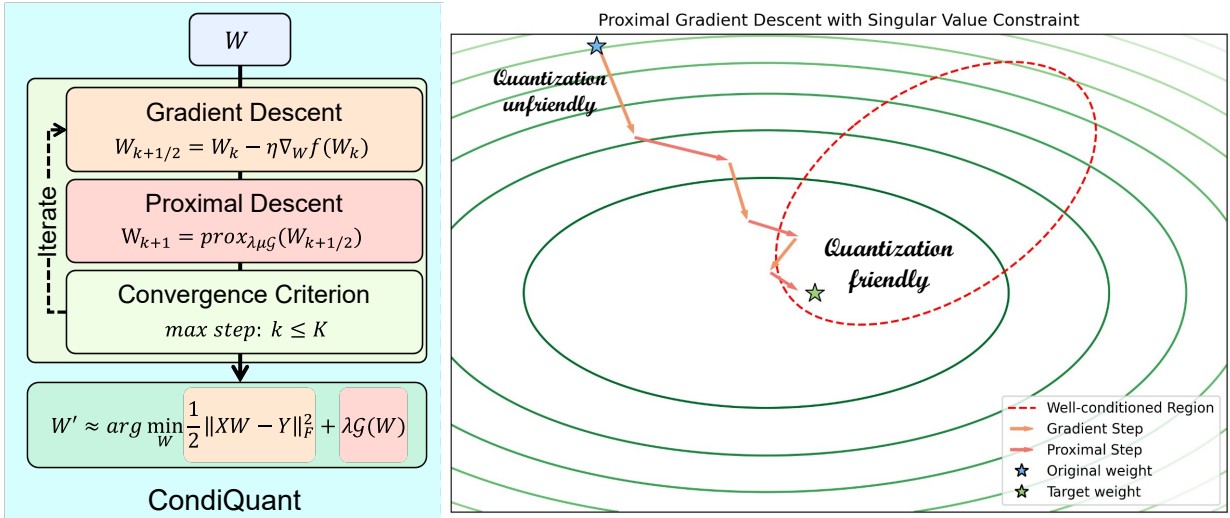

*Figure 3.* Overview of our proposed CondiQuant, which employs iterative optimization to minimize the condition number of the weight matrix while maintaining the output. The gradient descent step updates the weight matrix with gradients to ensure the output is close to the original. The proximal descent step minimizes the condition number with the proximal operator. The above two steps are conducted iteratively before reaching the convergence criterion, *i.e.,* the max iteration step. We illustrate the effect of both steps. With CondiQuant, the weight matrix is converted into a quantization-friendly and well-conditioned one, while the output is close to the original.

where $\mathcal{G}(\cdot)$ is a regularization term used to compact the singular value distribution and $\lambda$ is the regularization strength parameter. There are several forms of $\mathcal{G}(W)$ to compact the singular values' distribution. Considering efficiency and effect, we choose the following form:

$$\mathcal{G}(W) := \sum_{i=1}^{r} (\sigma_i(W) - t)^2, \tag{10}$$

where $r = rank(W)$ represents the number of singular values and $t$ is a value between $\sigma_{min}(W)$ and $\sigma_{max}(W)$. The median or the mean of the singular values are possible candidates for $t$ and we choose the mean value form in CondiQuant. With this form, we could provide a closed solution to minimize $\mathcal{G}(W)$ in the following section.

**Proximal Operator for $\mathcal{G}(W)$.** We utilize the proximal operator to minimize the $\mathcal{G}(W)$ part. We define the proximal operator of function $\mathcal{G}(W)$ as follows:

$$\text{prox}_{\lambda\mu\mathcal{G}}(W) := \arg\min_{Z} \left\{ \frac{1}{2}\|Z - W\|_F^2 + \lambda\mu\mathcal{G}(Z) \right\}, \tag{11}$$

where $\mu$ is a hyper-parameter to balance the optimization. The target of Eq. 11 is to minimize $\mathcal{G}(Z)$ and keep $Z$ close to $W$. To solve this problem, we perform singular value decomposition (SVD) on $W$ to extract the singular values:

$$W = U\Sigma_W V^T, \tag{12}$$

where $U$ and $V$ are orthogonal matrices and $\Sigma_W = \text{diag}(\sigma_i(V))$ is the diagonal matrix of singular values. The problem could be reduced to optimize each singular value $\sigma_i(W)$ independently and can be written as:

$$\sigma_i^*(W) := \arg\min_{\sigma_i^*} \left\{ \frac{1}{2}(\sigma_i^* - \sigma_i(W))^2 + \lambda\mu(\sigma_i^* - t)^2 \right\}. \tag{13}$$

We notice that Eq. 13 is a quadratic minimization problem,

and a closed solution could be derived. Specifically, after taking the derivative and setting it to zero, we have:

$$\sigma_i^*(W) = \frac{\sigma_i(W) + 2\lambda\mu t}{1 + 2\lambda\mu}. \tag{14}$$

Using the updated singular values $\sigma_i^*(W)$ , we reconstruct the matrix with lower condition number:

$$W^* = U\Sigma_{W^*}V^T, \quad \Sigma_{W^*} = \text{diag}(\sigma_i^*(W)). \tag{15}$$

To conclude, with the proximal operator, we could update $W$ to $W^*$ to minimize $\mathcal{G}(W)$. Hereby, the condition number of $W$ is reduced and the output is kept still.

**Proximal Gradient Descent.** As discussed previously, the optimization problem contains both smooth and non-smooth components. More specifically, $\frac{1}{2}\|XW - Y\|_F^2$ is a convex and differentiable function, while $\mathcal{G}(W)$ is a convex but non-differentiable function. With the proximal operator, we design the following proximal gradient descent method to solve two components iteratively:

**Step1. Gradient Descent.** Update the smooth part $\frac{1}{2}\|XW - Y\|_F^2$ with the gradient descent:

$$W_{k+\frac{1}{2}} = W_k - \eta \left. \nabla_W \frac{1}{2}\|XW - Y\|^2 \right|_{W=W_k}, \tag{16}$$

where $\eta > 0$ is the step size, and $\nabla_W$ is the gradient of $\frac{1}{2}\|XW - Y\|_F^2$ at $W_k$. With this step, we ensure the output is as consistent as possible with the original value.

**Step2. Proximal Step.** Apply the proximal operator to the non-smooth part $\mathcal{G}(W)$ with Eq. 12, 14 and 15 to minimize the condition number:

$$W_k = \text{prox}_{\lambda\mu\mathcal{G}}(W_{k+\frac{1}{2}}) = U\Sigma_{W^*}V^T.$$

**Step3. Repeat.** Iterate the above two steps until reaching the convergence criterion, *i.e.,* $k \geq K$. $k$ is the iteration step

*(a) Ablation study on learning rate $\eta$.*

| $\eta$ | $10^{-1}$ | $10^{-2}$ | $10^{-3}$ | $10^{-4}$ |
|---|---|---|---|---|
| PSNR↑ | 37.13 | 37.15 | 37.15 | 37.08 |
| SSIM↑ | 0.9568 | 0.9568 | 0.9567 | 0.9564 |

*(b) Ablation study on target value $t$.*

| $t$ | Mean | Median | $(\sigma_{max} + \sigma_{min})/2$ |
|---|---|---|---|
| PSNR↑ | 37.15 | 37.12 | 37.11 |
| SSIM↑ | 0.9567 | 0.9566 | 0.9567 |

*(c) Ablation study on $\lambda$.*

| $\lambda$ | 0.001 | 0.002 | 0.004 | 0.005 | 0.010 |
|---|---|---|---|---|---|
| PSNR↑ | 37.13 | 37.12 | 37.13 | 37.10 | 37.08 |
| SSIM↑ | 0.9567 | 0.9566 | 0.9567 | 0.9566 | 0.9565 |

*Table 2.* Ablation studies of CondiQuant on Set5 ($\times$2, 2 bits). The selection of hyperparameters and the effect of CondiQuant are evaluated. The results demonstrate that CondiQuant is robust and improves performance.

while $K$ is the total step of iteration. We select a desired $K$ such that the difference between consecutive iterates is sufficiently small while the iteration is not too long.

We illustrate our proposed CondiQuant in Fig. 3. With the proposed CondiQuant, we convert the weight metric from the quantization-unfriendly position into a friendly position. Besides, both gradient descent and proximal descent steps are calculation efficient and the cost is inexpensive. Moreover, we introduce no additional module. Therefore, no computation and storage overhead is caused during the inference stage and we obtain the theoretic optimal compression and speedup ratio in post-training quantization. After the conversion, we leverage the previous work's scheme (Liu et al., 2024) to perform quantization and calibration, and the details are described in the supplementary materials.

## 4. Experiments

To exhibit the outstanding performance of our proposed CondiQuant, comparisons with SOTA methods are also provided in both quantitative and qualitative forms. Extensive ablation studies are conducted to show the robustness and effectiveness of our elaborate designs.

### 4.1. Experimental Settings

**Data and Evaluation.** We employ DF2K (Timofte et al., 2017; Lim et al., 2017) as the calibration set, which combines DIV2K (Timofte et al., 2017) and Flickr2K (Lim et al., 2017). During calibration, we only use low-resolution ones and the FP model. Thereafter, CondiQuant is tested on five commonly used benchmarks in the SR field, including Set5 (Bevilacqua et al., 2012), Set14 (Zeyde et al., 2010), B100 (Martin et al., 2001), Urban100 (Huang et al., 2015), and Manga109 (Matsui et al., 2017). The evaluation metrics include PSNR and SSIM (Wang et al., 2004), which are calculated on the Y channel of the YCbCr space.

**Implementation Details.** The backbone of CondiQuant is SwinIR-light (Liang et al., 2021), a small and efficient ViT for image restoration. The scale factors include $\times$2, $\times$3, and $\times$4, and bit-width includes 2, 3, and 4 bits. During condition number optimization, the calibration set size is 100 images, randomly selected from DF2K and cropped to $3\times64\times64$. We set step size $\eta = 10^{-2}$, regularization coefficient $\lambda = 0.003$, $\mu = 1$, and max iteration $K = 50$. The target value $t$ is the mean value of singular values. Our code is written with PyTorch (Paszke et al., 2019) and runs on an NVIDIA A800-80GB GPU.

### 4.2. Ablation Study

In this section, we conduct four ablation studies on Set5 ($\times$2) with 2 bits to evaluate our design.

**Learning Rate $\eta$.** In CondiQuant, $\eta$ decides the step size of the gradient descent step. We evaluate $\eta$ from $10^{-4}$ to $10^{-1}$. As shown in Tab. 2a, the impact of $\eta$ is minor in the evaluation range. We attribute this minor impact to the rank deficiency across the network and the design of proximal descent. Rank deficiency allows multiple candidates for $W$ while the SVD reconstruction also guarantees the output remains the same. Therefore, the difference between $W_{k+\frac{1}{2}}$ and $W_k$ is small enough. However, the gradient step is not unnecessary as it ensures the output won't change much.

**Selection of $t$.** As discussed in Sec.3.3, there are several ways to select the target value $t$ in $\mathcal{G}(W)$, including the mean value and median value of all singular values and $(\sigma_{min} + \sigma_{max})/2$. We try these variants and the result is shown in Tab. 2b. The stable performance with different variants indicates that, as long as the singular values are compacted, the performance could therefore be improved. Therefore, $t$ is assigned with the mean singular values.

**Regularization Coefficient $\lambda$.** As shown in Tab. 2c, we test different $\lambda$ from 0.001 to 0.01. The optimal results could be obtained when $\lambda$ is set to 0.004. Other settings also provide good results, showing the robustness of CondiQuant.

### 4.3. Comparison with State-of-the-Art Methods

We adopt two kinds of SOTA PTQ methods for comparison. The first kind is PTQ methods specifically for SR, including DBDC+Pac (Tu et al., 2023), 2DQuant (Liu et al., 2024). The second kind includes PTQ4ViT (Yuan et al., 2022), RepQ (Li et al., 2023), and NoisyQuant (Liu et al., 2023), which are designed for general vision transformers.

**Quantitative Results.** Table 3 shows the comprehensive results comparing with edge PTQ methods with $2 \sim 4$ bits and $2 \sim 4$ scale factors on five common SR benchmarks. With two universally accepted metrics, our proposed CondiQuant outperforms all other methods on all benchmarks regardless of bit-width and scale factor. Specifically, CondiQuant has a huge improvement of 1.11 dB and 1.60 dB on $\times$4, 2-bit on Set5 and Manga109 datasets, respectively. Besides, the average difference between $\times$2, 4 bits CondiQuant, and the full-precision model is 0.38 dB. This means the quantization degradation when compressing to only 4 bits is minimal and small enough for real-world deployment on edge devices.

*Table 3.* Quantitative comparison with SOTA methods. The first and second highest methods are marked with **red** and **blue** respectively. Our proposed CondiQuant remarkably outperforms **all** other methods in **all** settings on **all** benchmarks.

| Method | Bit | Set5 (×2) PSNR↑ | SSIM↑ | Set14 (×2) PSNR↑ | SSIM↑ | B100 (×2) PSNR↑ | SSIM↑ | Urban100 (×2) PSNR↑ | SSIM↑ | Manga109 (×2) PSNR↑ | SSIM↑ |
|---|---|---|---|---|---|---|---|---|---|---|---|
| SwinIR-light (Liang et al., 2021) | 32 | 38.15 | 0.9611 | 33.86 | 0.9206 | 32.31 | 0.9012 | 32.76 | 0.9340 | 39.11 | 0.9781 |
| Bicubic | 32 | 32.25 | 0.9118 | 29.25 | 0.8406 | 28.68 | 0.8104 | 25.96 | 0.8088 | 29.17 | 0.9128 |
| DBDC+Pac (Tu et al., 2023) | 4 | 37.18 | 0.9550 | 32.86 | 0.9106 | 31.56 | 0.8908 | 30.66 | 0.9110 | 36.76 | 0.9692 |
| PTQ4ViT (Yuan et al., 2022) | 4 | 37.43 | 0.9571 | 33.19 | 0.9139 | 31.84 | 0.8950 | 31.54 | 0.9212 | 37.59 | 0.9735 |
| RepQ (Li et al., 2023) | 4 | 37.89 | 0.9600 | 33.47 | 0.9174 | 32.08 | 0.8975 | 31.98 | 0.9269 | 38.37 | 0.9763 |
| NoisyQuant (Liu et al., 2023) | 4 | 37.50 | 0.9570 | 33.06 | 0.9101 | 31.73 | 0.8936 | 31.31 | 0.9181 | 37.47 | 0.9723 |
| 2DQuant (Liu et al., 2024) | 4 | 37.87 | 0.9594 | 33.41 | 0.9161 | 32.02 | 0.8971 | 31.84 | 0.9251 | 38.31 | 0.9761 |
| CondiQuant (ours) | 4 | 38.03 | 0.9605 | 33.50 | 0.9180 | 32.16 | 0.8993 | 32.03 | 0.9282 | 38.57 | 0.9769 |
| DBDC+Pac (Tu et al., 2023) | 3 | 35.07 | 0.9350 | 31.52 | 0.8873 | 30.47 | 0.8665 | 28.44 | 0.8709 | 34.01 | 0.9487 |
| PTQ4ViT (Yuan et al., 2022) | 3 | 36.49 | 0.9510 | 32.49 | 0.9045 | 31.27 | 0.8854 | 30.16 | 0.9027 | 36.41 | 0.9673 |
| RepQ (Li et al., 2023) | 3 | 35.06 | 0.9325 | 31.29 | 0.8719 | 30.04 | 0.8512 | 29.17 | 0.8726 | 34.89 | 0.9518 |
| NoisyQuant (Liu et al., 2023) | 3 | 35.32 | 0.9334 | 31.88 | 0.8911 | 30.73 | 0.8710 | 29.28 | 0.8835 | 35.30 | 0.9537 |
| 2DQuant (Liu et al., 2024) | 3 | 37.32 | 0.9567 | 32.85 | 0.9106 | 31.60 | 0.8911 | 30.45 | 0.9086 | 37.24 | 0.9722 |
| CondiQuant (ours) | 3 | 37.77 | 0.9594 | 33.21 | 0.9151 | 31.94 | 0.8966 | 31.18 | 0.9197 | 38.01 | 0.9755 |
| DBDC+Pac (Tu et al., 2023) | 2 | 34.55 | 0.9386 | 31.12 | 0.8912 | 30.27 | 0.8706 | 27.63 | 0.8649 | 32.15 | 0.9467 |
| PTQ4ViT (Yuan et al., 2022) | 2 | 33.25 | 0.8923 | 30.22 | 0.8402 | 29.21 | 0.8066 | 27.31 | 0.8111 | 32.75 | 0.9093 |
| RepQ (Li et al., 2023) | 2 | 31.65 | 0.8327 | 29.19 | 0.7789 | 28.27 | 0.7414 | 26.56 | 0.7455 | 30.46 | 0.8268 |
| NoisyQuant (Liu et al., 2023) | 2 | 30.13 | 0.7620 | 28.80 | 0.7556 | 28.26 | 0.7421 | 26.68 | 0.7627 | 30.40 | 0.8204 |
| 2DQuant (Liu et al., 2024) | 2 | 36.00 | 0.9497 | 31.98 | 0.9012 | 30.91 | 0.8810 | 28.62 | 0.8819 | 34.40 | 0.9602 |
| CondiQuant (ours) | 2 | 37.15 | 0.9567 | 32.74 | 0.9103 | 31.55 | 0.8912 | 29.96 | 0.9047 | 36.63 | 0.9713 |

| Method | Bit | Set5 (×3) PSNR↑ | SSIM↑ | Set14 (×3) PSNR↑ | SSIM↑ | B100 (×3) PSNR↑ | SSIM↑ | Urban100 (×3) PSNR↑ | SSIM↑ | Manga109 (×3) PSNR↑ | SSIM↑ |
|---|---|---|---|---|---|---|---|---|---|---|---|
| SwinIR-light (Liang et al., 2021) | 32 | 34.63 | 0.9290 | 30.54 | 0.8464 | 29.20 | 0.8082 | 28.66 | 0.8624 | 33.99 | 0.9478 |
| Bicubic | 32 | 29.54 | 0.8516 | 27.04 | 0.7551 | 26.78 | 0.7187 | 24.00 | 0.7144 | 26.16 | 0.8384 |
| DBDC+Pac (Tu et al., 2023) | 4 | 33.42 | 0.9143 | 29.69 | 0.8261 | 28.51 | 0.7869 | 27.05 | 0.8217 | 31.89 | 0.9274 |
| PTQ4ViT (Yuan et al., 2022) | 4 | 33.77 | 0.9201 | 29.75 | 0.8272 | 28.62 | 0.7942 | 27.43 | 0.8361 | 32.50 | 0.9360 |
| RepQ (Li et al., 2023) | 4 | 34.08 | 0.9232 | 30.04 | 0.8345 | 28.88 | 0.8013 | 27.87 | 0.8462 | 32.98 | 0.9401 |
| NoisyQuant (Liu et al., 2023) | 4 | 33.13 | 0.9122 | 29.06 | 0.8093 | 27.93 | 0.7754 | 26.66 | 0.8143 | 31.94 | 0.9293 |
| 2DQuant (Liu et al., 2024) | 4 | 34.06 | 0.9231 | 30.12 | 0.8374 | 28.89 | 0.7988 | 27.69 | 0.8405 | 32.88 | 0.9389 |
| CondiQuant (ours) | 4 | 34.32 | 0.9260 | 30.29 | 0.8417 | 29.05 | 0.8039 | 28.05 | 0.8506 | 33.23 | 0.9431 |
| DBDC+Pac (Tu et al., 2023) | 3 | 30.91 | 0.8445 | 28.02 | 0.7538 | 26.99 | 0.6937 | 25.10 | 0.7122 | 28.84 | 0.8403 |
| PTQ4ViT (Yuan et al., 2022) | 3 | 32.75 | 0.9028 | 29.14 | 0.8113 | 28.06 | 0.7712 | 26.43 | 0.8014 | 31.20 | 0.9131 |
| RepQ (Li et al., 2023) | 3 | 31.04 | 0.8548 | 28.04 | 0.7572 | 26.83 | 0.7019 | 25.56 | 0.7493 | 30.16 | 0.8904 |
| NoisyQuant (Liu et al., 2023) | 3 | 30.78 | 0.8511 | 27.94 | 0.7624 | 26.98 | 0.7153 | 25.43 | 0.7481 | 29.64 | 0.8792 |
| 2DQuant (Liu et al., 2024) | 3 | 33.24 | 0.9135 | 29.56 | 0.8255 | 28.50 | 0.7873 | 26.65 | 0.8116 | 31.46 | 0.9235 |
| CondiQuant (ours) | 3 | 33.92 | 0.9224 | 30.02 | 0.8367 | 28.84 | 0.7986 | 27.37 | 0.8356 | 32.48 | 0.9367 |
| DBDC+Pac (Tu et al., 2023) | 2 | 29.96 | 0.8254 | 27.53 | 0.7507 | 27.05 | 0.7136 | 24.57 | 0.7117 | 27.23 | 0.8213 |
| PTQ4ViT (Yuan et al., 2022) | 2 | 29.96 | 0.7901 | 27.36 | 0.7030 | 26.74 | 0.6590 | 24.56 | 0.6460 | 27.37 | 0.7390 |
| RepQ (Li et al., 2023) | 2 | 27.32 | 0.6478 | 25.63 | 0.5918 | 25.44 | 0.5652 | 23.42 | 0.5582 | 24.51 | 0.5721 |
| NoisyQuant (Liu et al., 2023) | 2 | 27.53 | 0.6641 | 25.77 | 0.5952 | 25.37 | 0.5613 | 23.59 | 0.5739 | 26.03 | 0.6632 |
| 2DQuant (Liu et al., 2024) | 2 | 31.62 | 0.8887 | 28.54 | 0.8038 | 27.85 | 0.7679 | 25.30 | 0.7685 | 28.46 | 0.8814 |
| CondiQuant (ours) | 2 | 33.00 | 0.9130 | 29.44 | 0.8253 | 28.45 | 0.7882 | 26.36 | 0.8080 | 30.88 | 0.9203 |

| Method | Bit | Set5 (×4) PSNR↑ | SSIM↑ | Set14 (×4) PSNR↑ | SSIM↑ | B100 (×4) PSNR↑ | SSIM↑ | Urban100 (×4) PSNR↑ | SSIM↑ | Manga109 (×4) PSNR↑ | SSIM↑ |
|---|---|---|---|---|---|---|---|---|---|---|---|
| SwinIR-light (Liang et al., 2021) | 32 | 32.45 | 0.8976 | 28.77 | 0.7858 | 27.69 | 0.7406 | 26.48 | 0.7980 | 30.92 | 0.9150 |
| Bicubic | 32 | 27.56 | 0.7896 | 25.51 | 0.6820 | 25.54 | 0.6466 | 22.68 | 0.6352 | 24.19 | 0.7670 |
| DBDC+Pac (Tu et al., 2023) | 4 | 30.74 | 0.8609 | 27.66 | 0.7526 | 26.97 | 0.7104 | 24.94 | 0.7369 | 28.52 | 0.8697 |
| PTQ4ViT (Yuan et al., 2022) | 4 | 31.49 | 0.8831 | 28.04 | 0.7680 | 27.20 | 0.7240 | 25.53 | 0.7660 | 29.52 | 0.8940 |
| RepQ (Li et al., 2023) | 4 | 31.77 | 0.8880 | 28.32 | 0.7750 | 27.40 | 0.7310 | 25.83 | 0.7780 | 29.88 | 0.9010 |
| NoisyQuant (Liu et al., 2023) | 4 | 31.09 | 0.8751 | 27.75 | 0.7591 | 26.91 | 0.7151 | 25.07 | 0.7500 | 28.96 | 0.8820 |
| 2DQuant (Liu et al., 2024) | 4 | 31.77 | 0.8867 | 28.30 | 0.7733 | 27.37 | 0.7278 | 25.71 | 0.7712 | 29.71 | 0.8972 |
| CondiQuant (ours) | 4 | 32.09 | 0.8923 | 28.50 | 0.7792 | 27.52 | 0.7345 | 25.97 | 0.7831 | 30.16 | 0.9054 |
| DBDC+Pac (Tu et al., 2023) | 3 | 27.91 | 0.7250 | 25.86 | 0.6451 | 25.65 | 0.6239 | 23.45 | 0.6249 | 26.03 | 0.7321 |
| PTQ4ViT (Yuan et al., 2022) | 3 | 29.77 | 0.8337 | 27.00 | 0.7248 | 26.21 | 0.6735 | 24.22 | 0.6983 | 27.94 | 0.8479 |
| RepQ (Li et al., 2023) | 3 | 27.52 | 0.7419 | 24.84 | 0.5996 | 23.99 | 0.5351 | 22.42 | 0.5739 | 26.58 | 0.7838 |
| NoisyQuant (Liu et al., 2023) | 3 | 28.90 | 0.7972 | 26.50 | 0.6970 | 26.16 | 0.6628 | 23.86 | 0.6667 | 27.17 | 0.8116 |
| 2DQuant (Liu et al., 2024) | 3 | 30.90 | 0.8704 | 27.75 | 0.7571 | 26.99 | 0.7126 | 24.85 | 0.7355 | 28.21 | 0.8683 |
| CondiQuant (ours) | 3 | 31.62 | 0.8855 | 28.20 | 0.7715 | 27.31 | 0.7269 | 25.39 | 0.7624 | 29.29 | 0.8915 |
| DBDC+Pac (Tu et al., 2023) | 2 | 25.01 | 0.5554 | 23.82 | 0.4995 | 23.64 | 0.4544 | 21.84 | 0.4631 | 23.63 | 0.5854 |
| PTQ4ViT (Yuan et al., 2022) | 2 | 27.23 | 0.6702 | 25.38 | 0.5914 | 25.15 | 0.5621 | 22.94 | 0.5587 | 24.66 | 0.6132 |
| RepQ (Li et al., 2023) | 2 | 25.55 | 0.5834 | 23.54 | 0.4751 | 23.30 | 0.4298 | 21.62 | 0.4493 | 23.60 | 0.5561 |
| NoisyQuant (Liu et al., 2023) | 2 | 25.94 | 0.5862 | 24.33 | 0.5067 | 24.16 | 0.4718 | 22.32 | 0.4841 | 23.82 | 0.5403 |
| 2DQuant (Liu et al., 2024) | 2 | 29.53 | 0.8372 | 26.86 | 0.7322 | 26.46 | 0.6927 | 23.84 | 0.6912 | 26.07 | 0.8163 |
| CondiQuant (ours) | 2 | 30.64 | 0.8671 | 27.59 | 0.7567 | 26.93 | 0.7136 | 24.54 | 0.7282 | 27.67 | 0.8613 |

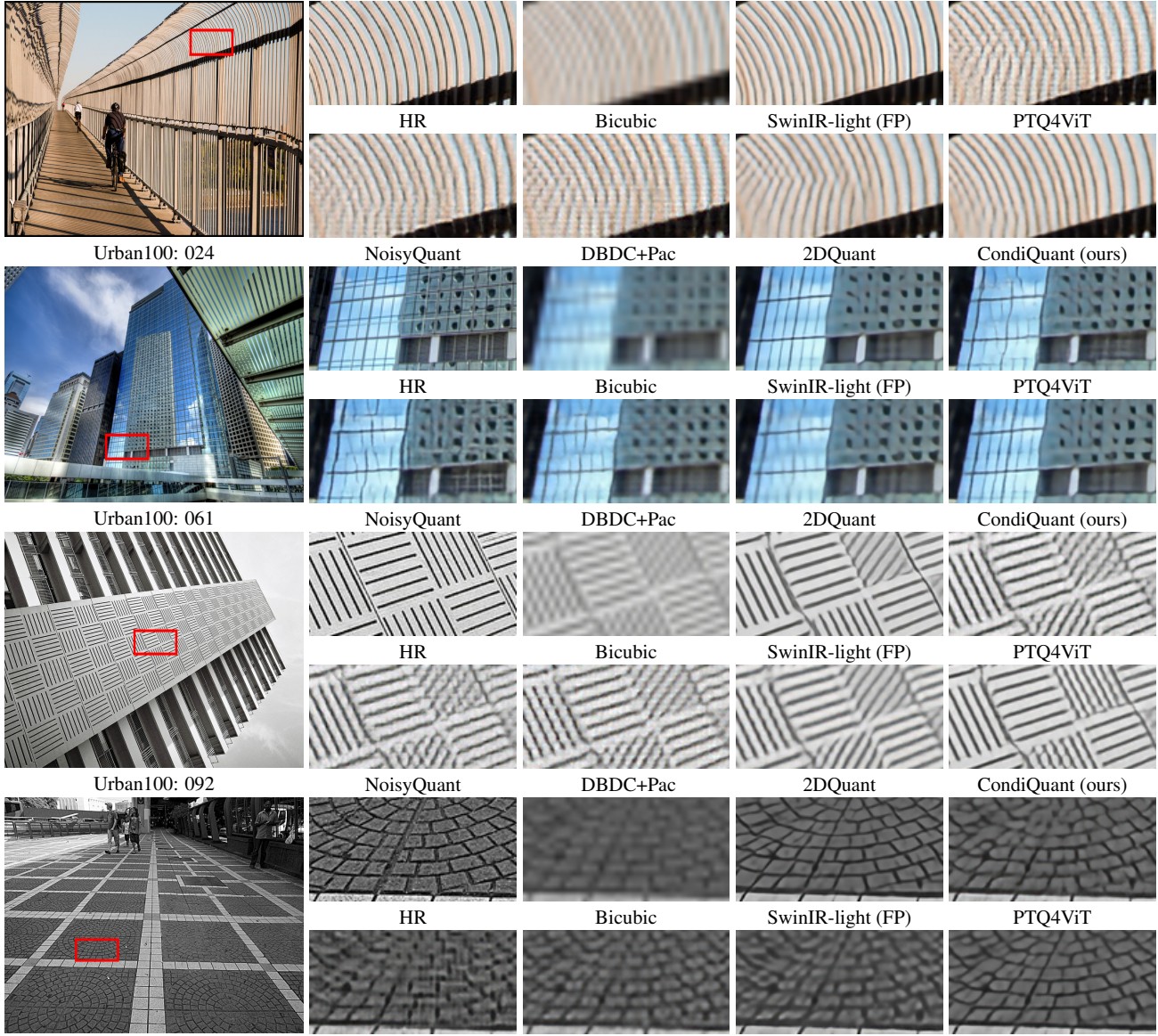

*Figure 4.* Visual comparison for image SR. We compare our proposed CondiQuant with current competitive quantization methods and the full-precision (FP) model. The visual results illustrate that CondiQuant gains sharper edges and reasonable textures.

With condition number optimization, the model's sensitivity to the quantization error is reduced. Thereafter, the weight matrix is efficiently converted into a more quantization-friendly representation. Furthermore, the quantized model enjoys significant improvement during the following calibration and distillation stage. Besides, the oscillation during distillation is reduced, and detailed supporting information is in the supplemental material.

**Visual Results.** Regularly occurring dense and fine textures are **typical challenging scenarios** for image super-resolution. As shown in Fig. 4, we choose the most challenging cases to validate the visual effect of CondiQuant with ×4. The competing methods are struggling to recover in-erratic textures and often generate artifacts. On the contrary,

with our proposed CondiQuant, the quantized model could reconstruct HR with rich details and reasonable structures. In 024 and 061, the direction of the texture is distorted, and jagged edges are generated with Bicubic. Given the misleading input, other methods can not provide robust restoration like the FP model. However, our proposed CondiQuant can still guarantee fidelity to the greatest extent.

Even more, in 092, the FP model can not provide results with accurate direction on the wall, while CondiQuant can correct the error and provide sharp contents. This is because the FP model suffers from over-fitting, and CondiQuant could ease this phenomenon. Besides, in 095, the competing methods recover bricks with vague texture while CondiQuant remains aligned with the FP model. In conclusion, compared to other

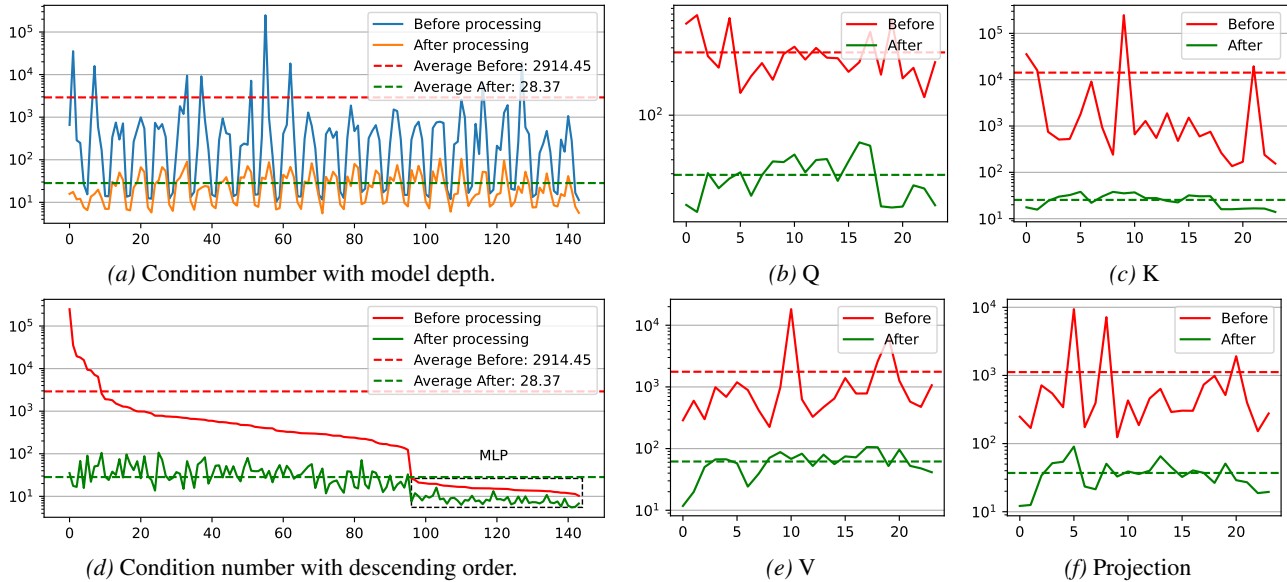

*Figure 5.* Condition number before and after CondiQuant on ×2 model. Before CondiQuant, the distribution of condition numbers is extremely high, with an average of nearly 3k. After CondiQuant, the average value of condition numbers is significantly reduced to 28.37.

*Table 4.* Performance on Manga109 (×4) with 4 bits.

|  | SwinIR | | MambaIRv2 | | EDSR | |
|---|---|---|---|---|---|---|
|  | PSNR | SSIM | PSNR | SSIM | PSNR | SSIM |
| FP | 30.92 | 0.9150 | 31.24 | 0.9182 | 30.35 | 0.9070 |
| 2DQuant | 29.71 | 0.8972 | 28.30 | 0.8540 | 28.95 | 0.8143 |
| CondiQuant | 30.16 | 0.9054 | 29.75 | 0.9007 | 29.24 | 0.8575 |

methods, CondiQuant achieves the minimum information loss after low bit quantization and restores images with high-fidelity, rich details, and sharper edges.

### 4.4. Condition Number Analysis

The excellent results are closely related to the decrease in condition number. Therefore, we visualize the condition number shifts in Fig. 5 on the SR model with the scale factor of 2. Figures 5a and 5d show the condition number of all layers (including Q, K, V, projection, FC1, and FC2) with different sequences. Fig. 5a is about model depth, and Fig. 5d sorts the condition number before CondiQuant with descending order. The other four figures show different layers with model depth as the X-axis.

Overall, the average condition number is hugely reduced from nearly 3k to merely 28.37. This evident decrease shows the effectiveness of the proximal descent step in CondiQuant. To be specific, a regular pattern is observed with model depth. The condition numbers of the FC1 and FC2 in the MLP layer are distinctly smaller than those of other layers, and their influence is also minor. Hence, considering efficiency, we do not perform CondiQuant on FC1 and FC2. The condition number of the K matrix in self-attention is usually extremely large. This is also consistent with the claim that the quantization degradation of ViT is mainly attributed to self-attention. Moreover, the distribution of the condition number after CondiQuant is greater in the middle

and smaller at both ends. This indicates that the beginning and end modules are more important and sensitive in image restoration. To conclude, CondiQuant significantly reduces the condition number with high efficiency.

### 4.5. Architecture Generalization

To validate the generalization on different architectures, the performance of CondiQuant on three different architectures is tested on Manga109 with 4bit at a scale factor of 4. Three representative architectures are selected. SwinIR (Liang et al., 2021) is for vision transformer, MambaIRv2 (Guo et al., 2025) is for Mamba, and EDSR (Lim et al., 2017) is for CNN. The consistently higher metrics represent the generalization ability across different architectures.

## 5. Conclusion

In this paper, we propose CondiQuant, a condition number based PTQ method for image super-resolution. We analyze that the degradation of quantization is attributed to the quantization of activation and leverage condition number to minimize the impact of the activation quantization error. Thereafter, we formulate the optimization problem to minimize the condition number while maintaining the output as it is. To optimize, we design the gradient descent step to keep the output still and the proximal descent step to reduce the condition number. Both steps are calculation-efficient and the entire iteration process takes only 19.0 seconds. As there's no additional module, we reach the theoretically optimal compression and speedup ratio. Specifically, when quantized to 2 bits, the compression ratio is 3.60× and the speedup ratio is 5.08×. The comparison experiments demonstrate the excellent performance of CondiQuant while the ablation studies present its robustness.

## Impact Statement

This paper presents work whose goal is to advance the field of Machine Learning. There are many potential societal consequences of our work, none which we feel must be specifically highlighted here.

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

## A. A conclusion of the motivation

It is a necessary step to perform model quantization on the SR networks before deployment. However, the quantization process leads to performance degradation. The degradation is mainly from the activation, as the features change according to different input images. However, the quantization error of activation is difficult to minimize, as it is strongly correlated with the input.

Therefore, we decided to minimize the **impact** of the quantization error of activation (not the error itself), which is represented by the condition number of the weight matrix. When the condition number is lower, the impact is smaller, even when the activation quantization error remains unchanged. As most of the activation is rank-deficient, we can shift the weight matrix without introducing additional error. Therefore, we propose CondiQuant to reduce the condition number of the weight matrix while keeping the output unchanged.

## B. Derivation of proximal gradient descent

In convex optimization problems, for differentiable objective functions, we can iteratively find the optimal solutions simply using gradient descent. For non-differentiable objective functions, we can still iterate toward the optimal solution by introducing subgradients. However, subgradient methods are much slower compared to gradient descent ones. To address this, for functions that are non-differentiable in their entirety but can be separated into differentiable and non-differentiable components, we can use a faster algorithm known as the proximal gradient descent method. Generally, consider an objective function that can be decomposed into two functions of the following forms:

$$f(x) = g(x) + h(x), \tag{17}$$

where $g(x)$ is a convex and differentiable function, while $h(x)$ is also convex but non-differentiable function. Using proximal gradient descent, a convergence rate of $O(1/\epsilon)$ can be achieved, where $\epsilon = f(x^k) - f(x^*)$ represents the deviation between the current iteration result and the optimal solution. Moreover, by accelerating the proximal gradient, a convergence rate of $O(1/\sqrt{\epsilon})$ can be attained.

**General form.** For the differentiable part $g(x)$, assuming the starting point is at $x_0$, the second-order Taylor expansion can be expressed as:

$$g(x) \approx g(x_0) + \nabla g(x_0)^T (x - x_0) + \frac{1}{2} \nabla^2 g(x_0)(x - x_0)^2. \tag{18}$$

Assuming $g(x)$ satisfies $L-$Lipschitz continuity, where the upper bound of the Hessian matrix is $L$. By setting the step size $\eta = \frac{1}{L}$, the second-order term can be simplified to $\frac{1}{2\eta}$. The problem can be written as:

$$\min_x \frac{1}{2\eta}[(x - x_0)^2 + 2\eta\nabla g(x_0)(x - x_0) + (\eta\nabla g(x_0))^2] + h(x)$$

$$= \min_x \frac{1}{2\eta}[x - x_0 + t\nabla g(x_0)]^2 + h(x)$$

$$= \min_x \frac{1}{2\eta}\|x - (x_0 - t\nabla g(x_0))\|_2^2 + h(x)$$

$$= \min_x \frac{1}{2\eta}\|x - z\|_2^2 + h(x),$$

where $z = x_0 - t\nabla g(x_0)$ is the form of gradient descent for $g(x)$. Therefore, its solution is:

$$x = \arg\min_x \frac{1}{2}\|x - z\|_2^2 + \eta g(x), \qquad (19)$$

Eq. 19 is also known as the proximal operator, denoted as:

$$\text{prox}_{\eta,g}(z) = \arg\min_x \frac{1}{2}\|x - z\|_2^2 + \eta \cdot h(x). \qquad (20)$$

Using Proximal Functions, we can define an iterative process for optimization, called the proximal descent method. First, choose an initial point $x_0$ and then iterate:

$$x_i = \text{prox}_{h,\eta}(x_{i-1} - \eta_i\nabla g(x_{i-1})), \quad i = 1, 2, 3, \ldots \qquad (21)$$

There are some advantages of using this method:

1. For many $h$ functions, the proximal projection $\text{prox}_{h,\eta}$ has an analytical solution.

2. The $\text{prox}_{h,\eta}$ depends only on $h$, so it can be used for different functions $g$.

3. $g$ can be any convex function, as long as we can compute its gradient.

---

**1 CondiQuant pipeline**

---

**Input:** Pretrained weight matrix $W$; Activation $X$ from calibration set; Learning rate or step size $\eta$; Penalty parameter $\lambda$; Maximum iterations $K$.

**Output:** Quantization friendly weight matrix $W'$.

1   $U, \Sigma, V \leftarrow \text{SVD}(W)$
2   $Y \leftarrow XW$
3 **for** $k = 1$ **to** $K$ **do**
4     $W_{k+\frac{1}{2}} \leftarrow W_k - \eta\nabla_W \frac{1}{2}\|Y - XW\|_F^2\Big|_{W=W_k}$
5     $U_k, \Sigma_k, V_k \leftarrow \text{SVD}(W_k)$
6     $t \leftarrow \text{mean}(W_k)$
7     $\Sigma_k^* \leftarrow (\Sigma_k + 2\lambda\mu t)/(1 + 2\lambda\mu)$
8     $W_{k+1} \leftarrow U_k\Sigma_k^*V_k$
9     $k \leftarrow k + 1$
10 **end for**
11 $W' \leftarrow W_K$
12 **return** $W'$

---

**Accelerated proximal gradient method.** Choose an initial point $x_{-1} = x_0$, and then repeat the following steps:

1. Use the result from the previous step as the momentum and calculate $z$:
$$z = x_{k-1} + \frac{k-2}{k+1}(x_{k-1} - x_{k-2}), \qquad (22)$$

2. Update $x$ using $z$:
$$x_k = \text{prox}_{\eta_k}(z - \eta_k\nabla g(z)), \qquad (23)$$

The first step of $k = 1$ is the standard proximal gradient update. Afterward, we use $z$ to replace $x_{k-1}$, transferring momentum from previous iterations to update the next gradient direction, making it align more closely with the current gradient.

## C. Calibration details

**Implementation details.** We adopted a per-channel granularity and quantize all the linear layers within the transformer blocks. Although some layers are full-rank, we still perform CondiQuant, given their limited impact. We made some modifications to the DOBI(Distribution-oriented bound initialization) algorithm of 2DQuant (Liu et al., 2024) simultaneously. Specifically, for the long-tail distribution activations, we narrow the search range to 10 evenly spaced numbers between the 90th percentile and the maximum value. For weights and activations that follow an approximately symmetric distribution, we will search both the upper and lower bounds. The upper bound search range is from the 90th percentile to the maximum value, while the lower bound range is from the minimum value to the 10th percentile. We will also search 10 values within each range.

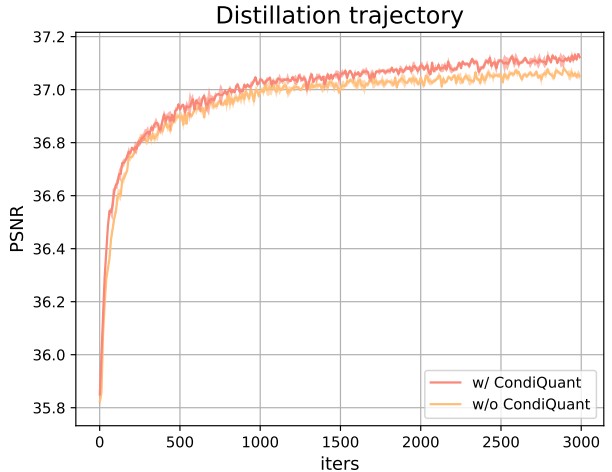
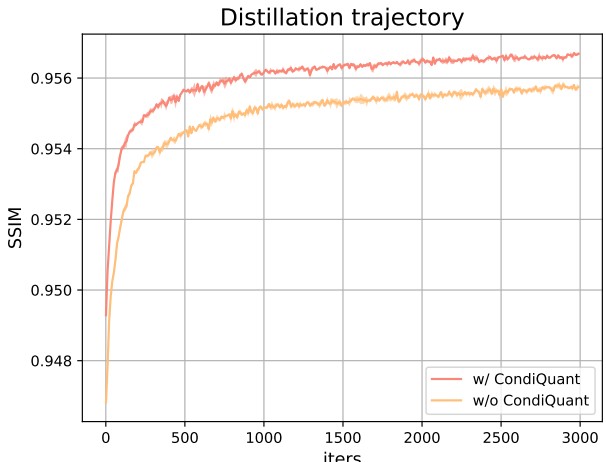

*(a)* Comparison of distillation trajectories of PSNR on Set5.     *(b)* Comparison of distillation trajectories of SSIM on Set5.

*Figure 6.* Condition number before and after CondiQuant on ×2 model. Before CondiQuant, the distribution of condition numbers is extremely high, with an average of nearly 3k. After CondiQuant, the average value of condition numbers is significantly reduced to 28.37.

**Visualization of distillation.** We provide more visual comparison results in knowledge distillation to demonstrate the role of CondiQuant. It is performed based on 2-bit quantization of SwinIR-light x4. We changed different random seeds for multiple distillations and obtained the distillation trajectory with a 95% confidence interval as shown in Fig. 6a and Fig. 6b.

## D. The ignored second-order term

In the main paper, we decompose the quantization error in the following form

$$
\begin{aligned}
\hat{Y} := \hat{X}\hat{W} &= (X + \delta X)(W + \delta W) \\
&= XW + X\delta W + \delta XW + \delta X\delta W.
\end{aligned}
\tag{24}
$$

The second-order $\delta X\delta W$ is directly ignored because the impact is minimal even when the bit-width is low. With 2 bits, for example, we have $|\delta W|_F \leq \frac{1}{4}|W|_F$. Such that we informally have

$$
|\delta X\delta W|_F^2 \approx |\delta X|_F^2|\delta W|_F^2 \ll |\delta X|_F^2|W|_F^2 \approx |\delta XW|_F^2.
\tag{25}
$$

With Eq (25), the second-order term is much smaller than the first-order term, meaning that the second-order term can be safely ignored.

## E. More visual results

We provide more visual comparison results in Figures 7 and 8. All results demonstrate the outstanding performance of our proposed CondiQuant.

---

**2** Modified DOBI pipeline

---

**Input:** Data to be quantized $v$, bit $b$, number of search point $K$.

**Output:** Clip bound $l$, $u$

1   $l \leftarrow \min(v)$

2   $u \leftarrow \max(v)$

3   $min\_mse \leftarrow +\infty$

4   **if** $v$ *is symmetrical* **then**

5     $v_{10} \leftarrow$ 10th percentile of $v$

6     $\Delta l \leftarrow (v_{10} - \min(v))/K$

7   **else**

8     $\Delta l \leftarrow 0$

9   **end if**

10   $v_{90} \leftarrow$ 90th percentile of $v$

11   $\Delta u \leftarrow (\max(v) - v_{90})/K$

12   **for** $i = 1$ *to* $K$ **do**

13     $l_i \leftarrow l + i \times \Delta l$

14     $u_i \leftarrow u - i \times \Delta u$

15     get $v_q$ based on quantization formula.

16     $mse \leftarrow \|v - v_q\|_2$

17     **if** $mse \leq min\_mse$ **then**

18       $min\_mse \leftarrow mse$

19       $l\_best \leftarrow l_i, u\_best \leftarrow u_i$

20     **end if**

21   **end for**

22   **return** $l\_best, u\_best$

---

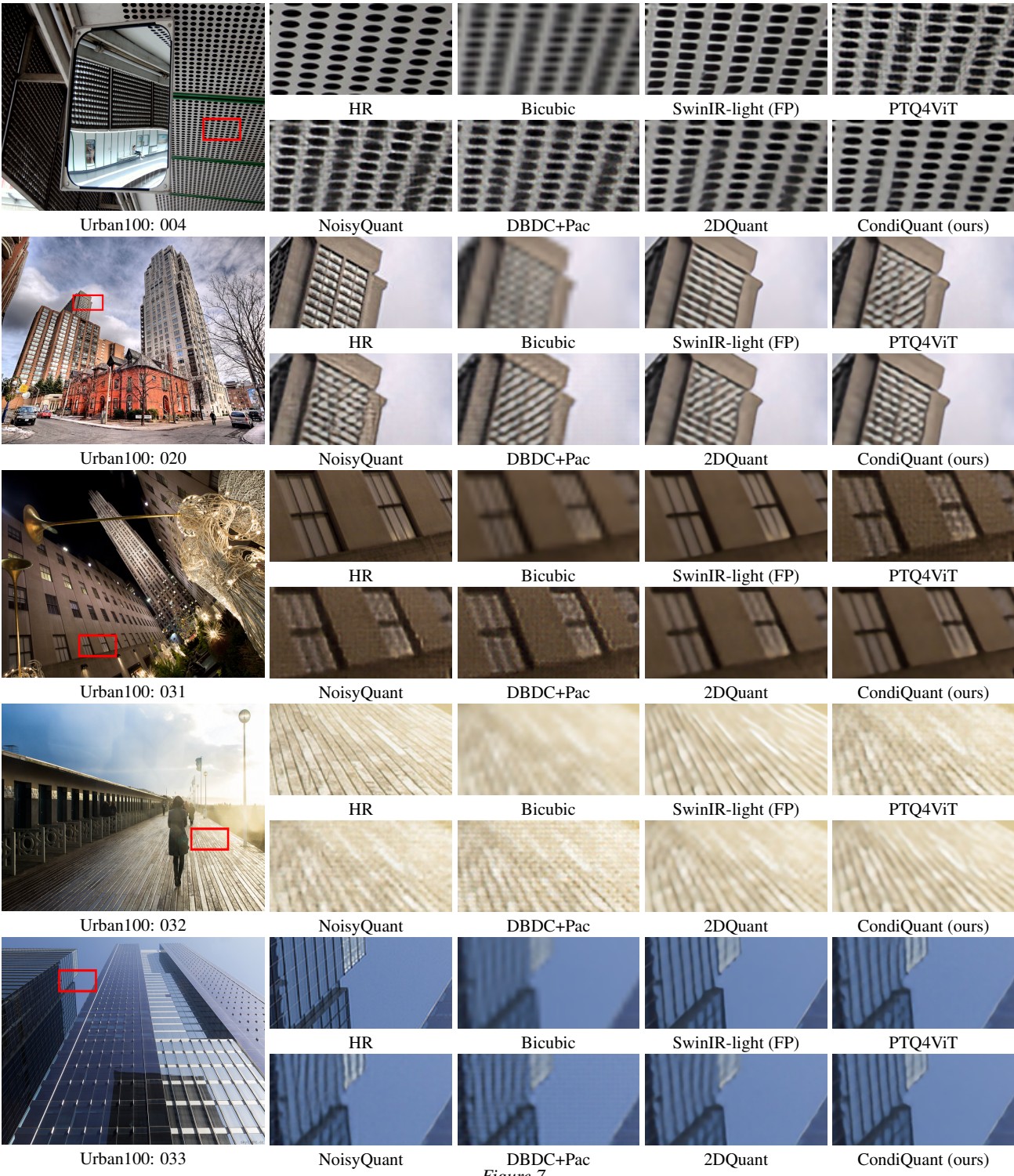

*Figure 7*

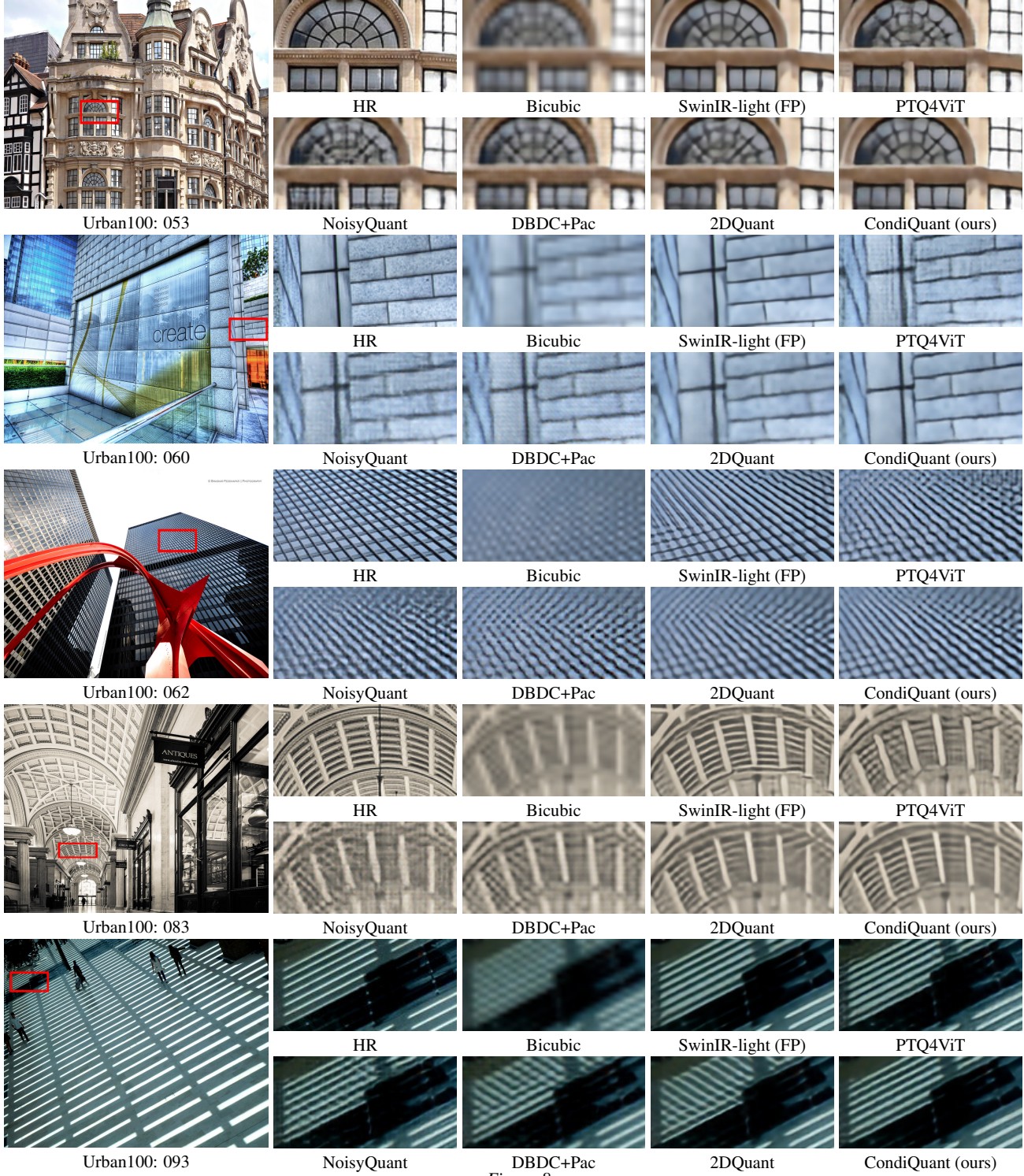

*Figure 8*

