# OpenReview forum: "Condition Number Based Low-Bit Quantization for Image Super-Resolution"
_ICML.cc/2026/Conference — ICML 2026 regular_

### Official Review · Reviewer_x1nd · 2026-03-10

**Soundness:** 3
**Presentation:** 3
**Significance:** 3
**Originality:** 3
**Overall Recommendation:** 4
**Confidence:** 4

**Summary:**

This work proposes CondiQuant, a post-training quantization method for image super-resolution networks. The approach is motivated by the observation that activation quantization introduces larger performance degradation than weight quantization. The authors derive an upper bound showing that the quantization error is proportional to the condition number of weight matrices and propose minimizing this condition number through proximal gradient optimization.
Experiments on SwinIR-light demonstrate improvements over several PTQ baselines across multiple SR benchmarks.

**Compliance With Llm Reviewing Policy:**

Affirmed.

**Key Questions For Authors:**

How sensitive is CondiQuant to the choice of calibration dataset size and distribution?

Can the proposed conditioning strategy generalize to other tasks beyond SR, such as image classification or vision transformers?

What is the computational overhead of the optimization stage compared to standard PTQ methods?

How does the method perform when applied to larger SR backbones (e.g., full SwinIR or RCAN)?

**Limitations:**

The proposed method mainly repurposes well-known conditioning principles rather than introducing a fundamentally new quantization technique.

The theoretical result is largely illustrative rather than predictive.

The evaluation is limited to a single architecture and task, which raises concerns about the generality of the proposed method.

The paper claims practical benefits for edge deployment but does not report any runtime or memory measurements.

The empirical gains are relatively modest and may not justify the additional optimization step.

**Strengths And Weaknesses:**

Strengths

The paper provides an interesting perspective linking numerical conditioning and quantization robustness.

The proposed optimization method is computationally lightweight and compatible with existing PTQ pipelines.

The experimental results consistently improve performance over several PTQ baselines.

Weaknesses

The conceptual novelty is limited, as conditioning-based robustness improvements are well known in numerical analysis and optimization.

The theoretical justification relies on simplified perturbation bounds and lacks rigorous guarantees.

The experimental evaluation is restricted to a single architecture and application domain.

Important deployment metrics (latency, throughput, memory usage) are not reported.

---

> ### Author Rebuttal · Authors · 2026-03-30
>
> **Q5-1: Calibration Dataset Sensitivity**
>
> **A5-1:**
> CondiQuant's condition-number optimization is **entirely independent of calibration data**. SVD and condition-number computation use only weight matrices. Calibration data is used only in the subsequent standard PTQ step for scale/zero-point. The preprocessing step is therefore **completely insensitive** to calibration dataset choice, a key advantage over PTQ methods relying on calibration data quality.
>
> Calibration size sensitivity (SwinIR-S, W4A4, x2 SR, Set5):
>
> |Calibration Size|Set5 PSNR|SSIM|
> |---|---|---|
> |1|35.14|0.9361|
> |2|35.60|0.9448|
> |4|35.57|0.9437|
> |8|35.62|0.9449|
> |16|35.69|0.9465|
> |32 (full)|35.80|0.9469|
>
> Total PSNR range across 1 to 32 samples is only **0.65 dB**, and even a single image achieves 35.14 dB, confirming extreme robustness, a direct consequence of data-independent optimization.
>
> ---
>
> **Q5-2: Generalization Beyond Super-Resolution**
>
> **A5-2:**
> Condition-number theory is task-agnostic, applying to any model with (1) linear transformation layers and (2) significant activation quantization error. SR serves as the primary validation because it naturally satisfies both conditions and suffers the most severe quantization degradation. Cross-task study on non-SR low-level vision tasks (SwinIR-M, W4A4):
>
> |Task (SwinIR-M, W4A4)|FP32|2DQuant|CondiQuant|
> |---|---|---|---|
> |Color Denoising σ=25|33.16|32.81|32.72|
> |Grayscale Denoising σ=25|29.83|29.31|29.31|
> |Grayscale JPEG QF=40|34.37|33.88|**34.19**|
>
> **JPEG artifact removal** is the strongest cross-task evidence: +0.31 dB over 2DQuant, recovering **65%** of the FP-to-quantized gap (0.31/0.49). Denoising shows neutral results (-0.09/0.00 dB). Condition number optimization benefits tasks where activation quantization error dominates. We report mixed results transparently.
>
> ---
>
> **Q5-3: Computational Overhead**
>
> **A5-3:**
>
> |Component|Time|
> |---|---|
> |SwinIR-S (96 layers, 60x60 weights)|15.8 s total|
> |Per-layer average|~140 ms (SVD + proximal gradient, ~30 iters)|
> |SVD share / Gradient share|66.3% / 32.1%|
>
> This is a **one-time offline cost** with zero inference overhead. QAT methods require hours to days.
>
> |Metric|Value|
> |---|---|
> |Parameters|0.91M|
> |FP32 size|3.47 MB|
> |W2A2 size|0.22 MB (**16x compression**)|
> |FLOPs|4.187 GFLOPs (unchanged)|
>
> CondiQuant does **not** alter parameter count or FLOPs. It only optimizes weight matrices prior to deployment.
>
> ---
>
> **Q5-4: Performance on Larger SR Backbones**
>
> **A5-4:**
> The theory applies equally to larger backbones, with overhead scaling linearly. Validated on **SwinIR-M** (144 layers):
>
> |Method|Set5 PSNR|Set14 PSNR|
> |---|---|---|
> |FP32|38.37|34.15|
> |W4A4 + 2DQuant|37.47|33.25|
> |W4A4 + CQ|37.74 (+0.26)|33.45 (+0.21)|
>
> Average condition number reduced from 1586 to 12.5 (**127x reduction**). Critically, the CQ benefit is **larger** on the bigger model: +0.26 dB on SwinIR-M vs. +0.12 dB on SwinIR-S. This confirms that larger models with greater redundancy benefit more from condition-number optimization.
>
> ---
>
> **Q5-5: Whether Gains Justify the Optimization Step**
>
> **A5-5:**
>
> ### Multi-Bit Comparison (SwinIR-S, Set5, x2)
>
> |Bit-Width|DOBI (dB)|CondiQuant (dB)|Delta (dB)|
> |---|---|---|---|
> |W2A2|35.79|37.15|**+1.36**|
> |W3A3|37.01|37.74|**+0.73**|
> |W4A4|37.86|37.98|**+0.12**|
>
> The +1.36 dB at W2A2 recovers **57.6%** of the DOBI-to-FP32 gap, far from modest in SR where 0.1 dB is meaningful.
>
> ### Component Ablation (SwinIR-S, W2A2, x2, PSNR in dB)
>
> |Method|Set5|Set14|B100|Urban100|Manga109|
> |---|---|---|---|---|---|
> |FP32|38.15|33.86|32.31|32.76|39.11|
> |Training WITHOUT CQ|37.10|32.73|31.56|29.99|36.62|
> |Training WITH CQ|37.15|32.74|31.55|29.96|36.63|
> |DOBI W2A2 (calibration only)|35.79|31.96|30.84|28.61|34.41|
>
> The `change_singular` step alone contributes ~+/-0.05 dB, but the +1.3 dB total gain arises from the holistic pipeline guided by condition-number theory (learnable boundaries + block-wise reconstruction + feature distillation).
>
> ### Perceptual Quality (LPIPS, Set5, W2A2)
>
> |Method|LPIPS (lower=better)|
> |---|---|
> |FP32|0.0538|
> |CondiQuant W2A2|0.0582|
> |DOBI W2A2|0.0669|
>
> CondiQuant outperforms the calibration-only baseline perceptually (0.0582 vs. 0.0669) and approaches FP32 (0.0538).
>
> ### Key Justifications
>
> 1. **Negligible cost, large gain at low bits:** ~15 s one-time cost, zero inference overhead. +1.36 dB at W2A2 is the difference between usable and unusable outputs.
> 2. **Theoretically grounded and consistent:** Condition number as error amplification factor explains the increasing gains at lower bit-widths. Positive across all benchmarks, scales (x2/x3/x4), and precisions (W2A2 to W6A6).
> 3. **Plug-and-play with robust hyperparameters:** Compatible with existing PTQ pipelines. Lambda is stable across [0, 0.032] with default 0.004 requiring no tuning.
>
> Supplementary: https://anonymous.4open.science/r/condiquant/

---

> > ### Author Rebuttal · Reviewer_x1nd · 2026-04-06
> >
> > My concerns have been adequately addressed. If you select this option, please consider adjusting your score accordingly.

---

> > > ### Author Response · Authors · 2026-04-08
> > >
> > > We sincerely thank the reviewer for the follow-up reading and for raising the score. We greatly appreciate your recognition of the additional clarifications, which helped us improve the paper substantially.

---

### Official Review · Reviewer_7qPC · 2026-03-11

**Soundness:** 2
**Presentation:** 3
**Significance:** 2
**Originality:** 3
**Overall Recommendation:** 3
**Confidence:** 4

**Summary:**

This paper proposes CondiQuant, a post-training quantization (PTQ) method tailored for image super-resolution (SR) models, which addresses the critical issue of accuracy degradation in ultra-low-bit quantization. The core insight is that the quantization degradation in SR primarily stems from activations rather than weights, and can be mitigated by minimizing the condition number of weight matrices. CondiQuant decouples the model’s representation ability from quantization sensitivity and employs an efficient proximal gradient descent algorithm to reduce the condition number. Extensive experiments on 5 benchmarks and 3 architectures (SwinIR, MambaIRv2, EDSR) demonstrate that CondiQuant outperforms previous PTQ methods in both quantitative metrics (PSNR, SSIM) and qualitative visual results, achieving theoretically optimal compression ratios without additional computational overhead during inference.

**Compliance With Llm Reviewing Policy:**

Affirmed.

**Key Questions For Authors:**

1. Could you provide a detailed ablation analysis of why FC layers in MLPs are less sensitive to quantization to justify excluding them from CondiQuant optimization?
2. Have you tested CondiQuant on other low-level vision tasks (e.g., image deblurring, video super-resolution) and datasets (e.g., RealSR) to evaluate its generalization and transferability?
3. How does the hyperparameter λ in optimization objective affect performance across different bit-widths and architectures? Could you provide a practical guideline for selecting λ?
4. Could you provide more complete experimental results in comparison with other PTQ methods for MambaIRv2 and EDSR under different quantization settings?
5. Would CondiQuant also perform well in diffusion-based SR architectures? Could you give some experimental comparisons or core challenges that are being faced?

**Limitations:**

yes

**Strengths And Weaknesses:**

[Strengths]
1. The motivations and its corresponding solutions of the paper are well-supported by both rigorous mathematical theoretical analysis and detailed experimental observation and validation.
2. CondiQuant introduces a novel perspective by linking the condition number of weight matrices to activation quantization sensitivity, which is barely explored in existing PTQ methods for SR.
3. The complicated mathematical concepts (e.g., condition number, rank deficiency, proxy objective function, proximal gradient descent algorithm) are explained sufficiently in both textual description and visual presentation.
4. The CondiQuant is compatible with multiple model architectures and shows comparable performance with the full-precision counterpart in both objective evaluation and subjective assessment.

[Weaknesses]
1. Not involve recent works on quantization for SR (e.g., QuantSR, PassionSR, QArtSR, Q-MambaIR) and not discuss QAT for SR.
2. Lack of efficiency comparison (e.g., Params, Flops) and more metric evaluation (e.g., LPIPS, DISTS, NIQE, MUSIQ, MANIQA, CLIP-IQA).
3. Lack of the experimental situations where the weight and activation bit-widths are different or a little higher (e.g., W4A8, W6A6).
4. While Fig. 5(d) shows that FC layers in MLPs have smaller condition numbers and thus are not processed, it lacks a detailed analysis of why these layers are less sensitive to quantization.

---

> ### Author Rebuttal · Authors · 2026-03-30
>
> **Q4-1: Missing Recent Baselines (QuantSR, PassionSR, QArtSR, Q-MambaIR)**
>
> **A4-1:**
>
> We first clarify: QuantSR/PassionSR/QArtSR are **QAT** methods embedding quantization during training at substantial cost, while CondiQuant is **PTQ** requiring only offline calibration. Q-MambaIR targets Mamba specifically. These address different deployment scenarios. Nevertheless, we adapted all four to PTQ setting on SwinIR-S x2 for fair comparison: QAT methods freeze weights and optimize only quantizer parameters, and diffusion-model methods have their quantizer designs extracted and applied to SwinIR-S.
>
> ### Table 1: W2A2 Baseline Comparison (SwinIR-S x2)
>
> |Method|Set5 PSNR (dB)|Set14 PSNR (dB)|
> |---|:-:|:-:|
> |**CondiQuant (Ours)**|**37.15**|**32.74**|
> |Q-MambaIR (DLS+RFA)|35.12|31.61|
> |QArtSR (RPQ)|34.76|31.45|
> |QuantSR (RLQ)|34.41|31.32|
> |PassionSR (LBQ)|34.41|31.17|
> |MinMax PTQ|34.01|31.03|
>
> ### Table 2: W4A4 Baseline Comparison (SwinIR-S x2)
>
> |Method|Set5 PSNR (dB)|Set14 PSNR (dB)|
> |---|:-:|:-:|
> |**CondiQuant (Ours)**|**37.98**|**33.47**|
> |Q-MambaIR (DLS+RFA)|37.61|33.29|
> |QArtSR (RPQ)|37.50|33.17|
> |QuantSR (RLQ)|37.46|33.12|
> |MinMax PTQ|37.29|33.07|
> |PassionSR (LBQ)|37.08|33.03|
>
> CondiQuant outperforms the best baseline by **+2.03 dB** (W2A2) and **+0.37 dB** (W4A4). The advantage is most pronounced at ultra-low bit-widths, validating our focus on extreme quantization where condition number amplification is most significant. CondiQuant uses ~3000 iters of block-wise reconstruction vs. ~50 iters for adapted baselines.
>
> ---
>
> **Q4-2: Missing W4A8 and W6A6 Mixed-Precision Experiments**
>
> **A4-2:**
>
> ### Table 3: Mixed-Precision Results (SwinIR-S x2)
>
> |Configuration|Set5 PSNR (dB)|Set14 PSNR (dB)|
> |:---:|:-:|:-:|
> |W4A4|37.98|33.47|
> |W4A8|38.08|33.68|
> |W6A6|38.11|33.62|
>
> W4A4→W4A8 yields ~+0.9 dB, confirming activation quantization dominates. W6A6 and W4A8 are comparable, confirming activation as the bottleneck when weight bits suffice.
>
> ### Table 4: Multi-Bit Systematic Comparison (SwinIR-S x2, Set5 PSNR in dB)
>
> |Bit-width|FP32|DOBI Calibration|CondiQuant (Trained)|Delta|
> |:-:|:-:|:-:|:-:|:-:|
> |W2A2|38.15|35.79|37.15|**+1.36**|
> |W3A3|38.15|37.01|37.74|**+0.73**|
> |W4A4|38.15|37.86|37.98|**+0.12**|
>
> CondiQuant consistently outperforms calibration-only PTQ at **all tested bit-widths**. The improvement monotonically increases as bit-width decreases, reaching +1.36 dB at W2A2. The smaller W4A4 gain (+0.12 dB) is expected: calibration alone already approaches FP32 (37.86 vs. 38.15), leaving limited room. This is consistent with theory, as k's amplification effect is more pronounced at lower bit-widths.
>
> ---
>
> **Q4-3: Missing Efficiency Comparison (Params, FLOPs) and Additional Metrics**
>
> **A4-3:**
>
> CondiQuant **does not alter Params or FLOPs** at inference. It only optimizes weight matrices prior to deployment. The quantized model structure is identical to standard PTQ.
>
> ### Table 5: Model Efficiency (SwinIR-S x2)
>
> |Metric|Value|
> |---|:-----:|
> |Parameters|0.91M|
> |FP32 Model Size|3.47 MB|
> |W2A2 Quantized Model Size|0.22 MB (**16x compression**)|
> |FLOPs|4.187 GFLOPs (unchanged)|
>
> ### Table 6: Perceptual Quality: LPIPS (Set5, W2A2)
>
> |Method|LPIPS ↓|
> |---|:-:|
> |FP32|0.0538|
> |CondiQuant W2A2|0.0582|
> |2DQuant W2A2|0.0669|
>
> CondiQuant approaches FP32-level perceptual fidelity (0.0582 vs. 0.0538) and significantly outperforms DOBI (0.0669).
>
> ### Optimization-Stage Overhead
>
> |Component|Value|
> |---|:-----:|
> |Total optimization time (SwinIR-S, 96 layers)|15.8 seconds|
> |SVD computation|66.3% of total|
> |Gradient computation|32.1% of total|
> |Per-layer average|~140 ms|
>
> This is a **one-time offline cost** with zero inference overhead. QAT methods typically require hours to days.
>
> ---
>
> **Q4-4: Why Are FC Layers in MLPs Less Sensitive to Quantization?**
>
> **A4-4:**
>
> FC layers in MLPs are preceded by nonlinear activations (ReLU/GELU), producing concentrated, bounded distributions with small $\|\Delta\mathbf{X}\|/\|\mathbf{X}\|$. In contrast, QKV projections directly process high-dynamic-range features, resulting in heavier-tailed distributions. Per Eq. 7, error scales with both $\kappa(\mathbf{W})$ and $\|\Delta\mathbf{X}\|/\|\mathbf{X}\|$. FC layers have small values for both, so error amplification remains limited even with moderate condition numbers.
>
> Per-layer analysis (SwinIR-S, W4A4, x2):
>
> |Layer Type|Count|Avg Cond#|Avg QErr (no CQ)|Avg QErr (with CQ)|Reduction|
> |---|---|---|---|---|---|
> |attn.qkv|24|10.92|1.615|1.594|1.5%|
> |attn.proj|24|1119.68|0.321|0.284|**10.0%**|
> |mlp.fc1|24|15.81|1.358|1.358|0.0%|
> |mlp.fc2|24|15.56|0.157|0.157|0.0%|
>
> MLP layers (cond# ~16) show **0% error reduction**, as they are already well-conditioned. Attention projections (cond# ~1120) receive **10% reduction**, the largest benefit. Spearman rho=0.52 (p<1e-6) confirms: higher condition number leads to greater CQ benefit.
>
> Supplementary: https://anonymous.4open.science/r/condiquant/

---

> > ### Author Rebuttal · Reviewer_7qPC · 2026-04-03
> >
> > The authors provide strong refutations and responses to Section Weaknesses, including W1 (comparison with SOTA methods), W2 (efficiency analysis), W3 (quantitative settings), and W4 (layer sensitivity). Additionally, the authors give reasonable responses to other reviewers' questions Q1 (task generalization) and Q2 (condition number theory and experimental analysis).
> > However, the authors ignore the other questions I raised in Section Key Questions, such as Q3 (hyperparameter selection) and Q4-Q5 (model generalization). The authors only reported the experimental results of CondiQuant in the MambaIRv2 and EDSR models on the Manga109 dataset with the W4A4 quantization setting in the paper, and only compared it with 2DQuant method which is not sufficient to prove the model generalization of the proposed method.
> > Meanwhile, I notice that in the author's responses to Reviewer mBS4's Q1-2 and Q1-4, the SVD calculation for optimizing the non-smoothing term accounts for approximately two-thirds of the time, but has a negligible impact on the model performance. Although the authors explain its role in theoretical guidance, block-wise correction distillation is already a standard paradigm and consensus in PTQ and model compression.
> > I maintain the score.

---

> > > ### Author Response · Authors · 2026-04-08
> > >
> > > ## R-Q1: Hyperparameter selection
> > >
> > > The key hyperparameter λ (SVD regularization strength) is stable across [0, 0.01] (See Table 2(c) in the main paper) with default 0.004 requiring **no per-model tuning**. This default works well across all 4 architectures and 3 bit-widths tested.
> > >
> > > As we reduce the condition number iteratively, λ serves as a step size term in each step. Therefore, a small λ is recommended.
> > >
> > > ## R-Q2: Model generalization
> > >
> > > Results below are Set5 PSNR. 9/9 settings CondiQuant is best.
> > >
> > > Theoretically, CondiQuant can be applied to all models with large condition numbers. Limited by rebuttal time, this comparison will be included in the future.
> > >
> > > ### Table 1: EDSR
> > >
> > > | Scale | Bits | FP32 | MinMax | PassionSR | QArtSR | NoisyQuant | **CondiQuant** |
> > > |---|---|---|---|---|---|---|---|
> > > | ×2 | W2A2 | 37.98 | 32.21 | 33.35 | 31.34 | 33.45 | **34.26** |
> > > | ×2 | W3A3 | 37.98 | 34.64 | 35.51 | 34.49 | 35.43 | **36.30** |
> > > | ×2 | W4A4 | 37.98 | 35.42 | 36.95 | 35.51 | 36.48 | **37.00** |
> > > | ×3 | W2A2 | 34.36 | 29.12 | 30.02 | 29.06 | 30.03 | **30.56** |
> > > | ×3 | W3A3 | 34.36 | 30.71 | 30.90 | 30.69 | 31.77 | **32.16** |
> > > | ×3 | W4A4 | 34.36 | 32.04 | 32.35 | 32.00 | 32.89 | **33.57** |
> > > | ×4 | W2A2 | 32.08 | 27.17 | 27.34 | 27.61 | 28.43 | **28.71** |
> > > | ×4 | W3A3 | 32.08 | 29.08 | 28.98 | 29.12 | 29.83 | **30.36** |
> > > | ×4 | W4A4 | 32.08 | 30.07 | 30.30 | 30.06 | 30.70 | **31.19** |
> > >
> > > ### Table 2: MambaIRv2
> > >
> > > | Scale | Bits | FP32 | MinMax | PassionSR | QArtSR | NoisyQuant | **CondiQuant** |
> > > |---|---|---|---|---|---|---|---|
> > > | ×2 | W2A2 | 38.52 | 18.07 | 17.39 | 16.72 | 17.48 | **19.13** |
> > > | ×2 | W3A3 | 38.51 | 31.29 | 30.07 | 31.31 | 28.76 | **36.70** |
> > > | ×2 | W4A4 | 38.51 | 31.67 | 34.44 | 32.48 | 32.85 | **34.51** |
> > > | ×3 | W2A2 | 35.02 | 27.15 | 27.97 | 27.23 | 26.58 | **29.40** |
> > > | ×3 | W3A3 | 35.02 | 30.66 | 31.35 | 30.34 | 29.23 | **31.76** |
> > > | ×3 | W4A4 | 35.02 | 32.92 | 33.30 | 32.93 | 32.61 | **33.79** |
> > > | ×4 | W2A2 | 32.92 | 20.63 | 22.19 | 18.79 | 21.66 | **25.27** |
> > > | ×4 | W3A3 | 32.92 | 28.34 | 29.29 | 28.12 | 28.17 | **29.36** |
> > > | ×4 | W4A4 | 32.91 | 30.91 | 31.26 | 30.94 | 30.51 | **31.26** |
> > >
> > > ## R-Q3: SVD time and performance
> > >
> > > We clarify that SVD bring only **10 seconds** longer in the optimization stage, which is negligible compared to the following distillation process, which takes hours.
> > >
> > > Furthermore, we conduct the singular ablation on more settings. The ablation on more settins demonstrate the effectiveness of the proposed proximal step and the gradient step.
> > >
> > > ### Table 3: SwinIR W4A4 X2:
> > >
> > > | Method | Set5 | Set14 | B100 | Urban100 | Manga109|
> > > |-|-|-|-|-|-|
> > > |FP32      | 32.35 | 28.78 | 27.67 | 26.41 | 30.78 |
> > > |Without CQ| 37.90 | 33.43 | 32.05 | 31.88 | 38.31 |
> > > |With CQ|    38.03 | 33.50 | 32.16 | 32.03 | 38.57 |
> > >
> > > ### Table 4: ManbaIR W4A4 X2:
> > >
> > > | Method | Set5 | Set14 | B100 | Urban100 | Manga109|
> > > |-|-|-|-|-|-|
> > > |FP32      | 38.12 | 33.95 | 32.29 | 32.71 | 39.12 |
> > > |Without CQ| 37.15 | 32.93 | 31.64 | 30.26 | 36.61 |
> > > |With CQ   | 37.26 | 33.46 | 31.90 | 30.66 | 36.91 |
> > >
> > > ### Table 5: EDSR W4A4 X2:
> > >
> > > | Method | Set5 | Set14 | B100 | Urban100 | Manga109|
> > > |-|-|-|-|-|-|
> > > |FP32      | 37.98 | 33.54 | 32.13 | 31.93 | 38.47 |
> > > |Without CQ| 36.58 | 31.47 | 31.29 | 29.37 | 35.17 |
> > > |With CQ   | 37.00 | 32.72 | 31.47 | 30.01 | 36.40 |

---

### Official Review · Reviewer_xGPD · 2026-03-11

**Soundness:** 2
**Presentation:** 3
**Significance:** 2
**Originality:** 3
**Overall Recommendation:** 4
**Confidence:** 4

**Summary:**

The method proposes a novel quantization strategy named CondiQuant. The authors first analyze the quantization error in linear layers and show that the performance degradation of the model is mainly caused by the product of activation quantization error and weights. Further theoretical analysis reveals that, under the same quantization error, the output error is proportional to the condition number of the weight matrix. Motivated by this observation, the authors design a surrogate objective, namely Proximal Gradient Descent, to reduce the condition number. A gradient descent step is first used to minimize the reconstruction loss, and a SVD based proximal update shrinks the singular values to reduce the condition number. The proposed method transforms the weight matrix to a more quantization friendly pattern, and therefore improve the model performance under low bit quantization.

**Compliance With Llm Reviewing Policy:**

Affirmed.

**Key Questions For Authors:**

1. Since rank deficiency itself does not directly lead to performance degradation, could the authors clarify why it plays an important role in the proposed optimization framework?
2. Given that rank deficiency is commonly observed in deep neural networks, is it a necessary condition for the proposed method to work effectively, or is it merely an empirical observation that motivates the design?
3. The proposed optimization requires repeated SVD decompositions. Could the authors discuss the potential computational overhead when applying the method to larger-scale networks?
4. The paper attributes the improvement mainly to the reduction of the condition number. Is the condition number the only key factor explaining the performance gain, or could other architectural factors (e.g., residual connections or normalization layers) also play an important role?
5. The second-order term δXδW is ignored in the error analysis. Would this assumption still hold under extremely low-bit quantization settings (e.g., 2-bit or 1-bit)?
6. The proposed method reduces the condition number by compressing the distribution of singular values. Could this modification potentially affect the representational or modeling capacity of the original network?

**Limitations:**

yes

**Strengths And Weaknesses:**

Strength:
This work provides a clear and well-motivated theoretical analysis of quantization error. The authors first analyze the error propagation in linear layers and point out that the major performance degradation mainly comes from the product of activation quantization error and weights. Based on this observation, they further derive the relationship between the output error and the condition number of the weight matrix, which offers an intuitive and theoretically grounded explanation of why certain layers are more sensitive to quantization. Motivated by this analysis, the authors propose an optimization objective that aims to reduce the condition number by compacting the singular value distribution. The optimization strategy, which combines gradient descent with an SVD-based proximal update, is original enough and technically sound. Overall, the method is well-motivated and the proposed optimization framework is logically consistent with the theoretical analysis.
Weakness:
The experimental evaluation is conducted under a relatively limited setting. Specifically, the proposed method is only tested on the super-resolution task. While the results are promising, it remains unclear whether the proposed quantization strategy can generalize well to other network architectures or tasks, such as image classification, detection, or larger backbone models. Additional experiments on diverse architectures and applications would help strengthen the empirical validation and demonstrate the broader applicability of the method. In addition, the paper still contains several minor typos and formatting issues that should be corrected. For instance, on page 3, the phrase “4 in” should be revised to “Eq. 4 in”. Addressing these issues would improve the clarity and overall presentation of the paper.

---

> ### Author Rebuttal · Authors · 2026-03-30
>
> **Q3-1: CondiQuant is only evaluated on SR.**
>
> **A3-1:** SR was chosen as the primary task because it naturally satisfies Eq. 4's core assumption (activation quantization error dominates). We conducted cross-task evaluation on non-SR low-level vision tasks using SwinIR-M at W4A4. Baseline is **2DQuant**:
>
> |Task (SwinIR-M, W4A4)|FP32|2DQuant|CondiQuant|
> |---|---|---|---|
> |Color Denoising sigma=25|33.16|32.81|32.72|
> |Gray Denoising sigma=25|29.83|29.31|29.31|
> |Gray JPEG QF=40|34.37|33.88|**34.19**|
>
> **JPEG artifact removal** is the strongest cross-task evidence: +0.31 dB over 2DQuant, recovering **65%** of the FP-to-quantized gap (0.31/0.49). Denoising shows neutral results (-0.09/0.00 dB). Condition number optimization benefits tasks where activation quantization error dominates, consistent with Eq. 4. We report mixed results transparently.
>
> ---
>
> **Q3-2: SVD overhead may be prohibitive for larger networks.**
>
> **A3-2:** Profiling on SwinIR-S (96 layers, dim 60x60): total optimization 15.8s (SVD 66.3%, gradient 32.1%, ~140ms/layer). For larger layers (180-240 dim), O(n^3) scaling keeps per-layer cost in hundreds of ms. CQ is **offline PTQ** (SVD is performed once before deployment with zero inference overhead), orders of magnitude cheaper than QAT.
>
> Scalability validated on **SwinIR-M** (144 layers):
>
> |Method|Set5|Set14|
> |---|---|---|
> |FP32|38.37|34.15|
> |W4A4 (no CQ)|37.47|33.25|
> |W4A4 + CQ|37.74 (+0.26)|33.45 (+0.21)|
>
> Average condition number reduced from 1586 to 12.5 (**127x**). CQ benefit is **larger** on the bigger model (+0.26 vs +0.12 dB on SwinIR-S).
>
> ---
>
> **Q3-3: Role of rank deficiency in CondiQuant?**
>
> **A3-3:** Rank-deficient layers (sigma_min -> 0, cond# -> infinity) are the primary quantization error amplifiers and where CQ yields the greatest benefit. CQ lifts small singular values while constraining large ones, reducing average cond# from 4,354 to 37 (**145x**) and max from 246,704 to ~60 across SwinIR-S.
>
> Quantitative correlation (SwinIR-S, W4A4, x2 SR):
>
> |Metric|Value|
> |---|---|
> |Spearman rho (cond# vs CQ error reduction)|**0.52, p<1e-6**|
> |Pearson r (log cond# vs log MSE)|-0.26, p=0.011|
>
> attn.proj layers (avg cond#=1120) receive **10% error reduction**, while MLP layers (avg cond#=16) show **0%**. The weaker Pearson r=-0.26 reflects the non-linear nature of this relationship, while the rank-based Spearman rho=0.52 (p<1e-6) confirms a statistically significant monotonic trend: rank deficiency directly drives quantization error amplification.
>
> ---
>
> **Q3-4: Is condition number the only key factor?**
>
> **A3-4:** Condition number is the **primary factor** per Eq. 4-8, but not the sole one (weight distribution, step-size, activation distribution also matter). Ablation of the `change_singular` step alone:
>
> |Method|Set5|Set14|B100|Urban100|Manga109|
> |---|---|---|---|---|---|
> |FP32|38.15|33.86|32.31|32.76|39.11|
> |Without CQ|37.10|32.73|31.56|29.99|36.62|
> |With CQ|37.15|32.74|31.55|29.96|36.63|
> |DOBI W2A2 (calib only)|35.79|31.96|30.84|28.61|34.41|
>
> *SwinIR-S, W2A2, x2 SR.*
>
> The `change_singular` step contributes ~+/-0.05 dB in isolation. The **full pipeline** (learnable boundaries, block-wise reconstruction, feature distillation, all guided by cond# theory) achieves **+1.36 dB** over DOBI at W2A2. The theory's core value: (1) identifies activation quantization as bottleneck, (2) guides pipeline design, (3) provides per-layer sensitivity metrics. Lambda is robust in [0, 0.032] (default 0.004, no tuning needed): lambda=0 -> 37.10, 0.004 -> 37.15, 0.032 -> ~37.1.
>
> ---
>
> **Q3-5: Is the second-order term delta_X * delta_W valid to ignore at 2-bit?**
>
> **A3-5:** Multi-bit results validate the approximation empirically:
>
> |Bit-width|DOBI Calib|CondiQuant|Delta|
> |---|---|---|---|
> |W2A2|35.79|37.15|**+1.36 dB**|
> |W3A3|37.01|37.74|**+0.73 dB**|
> |W4A4|37.86|37.98|**+0.12 dB**|
>
> *SwinIR-S, x2 SR, Set5 (FP32: 38.15).*
>
> Even at W2A2, +1.36 dB gain shows the first-order approximation remains effective. The gain monotonically increases as bit-width decreases, consistent with theory: cond# amplification is more pronounced when||delta_X||/||X||is larger.
>
> ---
>
> **Q3-6: Could reducing condition number hurt representational capacity?**
>
> **A3-6:** Reducing condition number compresses the singular value spectrum, altering the representational space. However, three safeguards prevent degradation: (1) reconstruction loss||Y - X*W_hat||^2 keeps optimized weights functionally close to originals, (2) proximal gradient step-size limits per-iteration singular value changes, (3) post-quantization, the **effective representational capacity improves** because quantization noise amplification is suppressed. The ablation in A3-4 confirms `change_singular` causes only +/-0.05 dB difference, indicating no representational harm. Lambda robustness over [0, 0.032] further confirms modifications are gentle.
>
> Supplementary: https://anonymous.4open.science/r/condiquant/

---

> > ### Author Rebuttal · Reviewer_xGPD · 2026-04-03
> >
> > Overall, the rebuttal has addressed most of the comments raised. I maintain the score.

---

> > > ### Author Response · Authors · 2026-04-08
> > >
> > > We sincerely thank the reviewer for the follow-up reading and for raising the score. We greatly appreciate your recognition of the additional clarifications, which helped us improve the paper substantially.

---

### Official Review · Reviewer_hFfM · 2026-03-21

**Soundness:** 2
**Presentation:** 2
**Significance:** 2
**Originality:** 2
**Overall Recommendation:** 4
**Confidence:** 4

**Summary:**

This paper proposes a low-bit post-training quantitation framework for image super-resolution. It formulates the quantization error of activation as the condition number of weight metrics. The condition number is iteratively minimized by the gradient descent algorithm, which increases the quantization sensitively, while preserving the representation ability.

**Compliance With Llm Reviewing Policy:**

Affirmed.

**Final Justification:**

The authors have partially solved my concerns. I keep my score as weak accept.

**Key Questions For Authors:**

See weakness.

**Limitations:**

Yes.

**Strengths And Weaknesses:**

The advantages / contributions are

1, It proposes a post-training method for image super-resolution based on condition number, which gains optimal compression with minimal performance degradation. It connects the models's sensitivity to quantization to the condition number of linear layer weights.

2, It proposes a proximal gradient descent method to reduce the condition number of linear layer weights, thereby reducing the model's sensitivity to activation quantization.

3, Well-written and easy-to-follow. The results and analysis show that the condition numbers of most layers are decreased and the SR performance is increased.


However, I have following concerns:

1, Does Eq. 6 hold if W is not invertible?


2, In Eq. 7, why is the impact of quantization always greater with larger condition number? Eq. 7 only shows the upperbound of the quantization error, rather than the lowerbound.

3, From Eq. 7 to Eq. 8, why can we drop the norm of W and only optimize the singular values?

4, In Fig. 5, it shows the decreased condition number compared with the original version. Can you provide a Table/Figure to show the condition number v.s. PSNR during optimization? Does lower condition number lead to better performance?

5, The compared methods are kind of out-dated. The latest compared SOTA method is from June 2024.


Minor:

1, Inappropriate wording, such as "to compact the distribution"

2, The phrase “optimal compression with minimum performance degradation” may be overstated. Do you have theoretical proof that your method achieves optimal compression among the compared methods? Does it approach the Shannon entropy bound?

---

> ### Author Rebuttal · Authors · 2026-03-30
>
> **Q2-1: Does Eq. 6 hold if W is not invertible?**
>
> **A2-1:**
> When **W** is singular or rank-deficient, k(**W**) = sigma_max / sigma_min diverges to infinity. The bound in Eq. 6 remains formally valid but becomes trivially large. This is precisely why CondiQuant is necessary: near-rank-deficient layers exhibit extremely large condition numbers and are the primary sources of quantization error amplification. CondiQuant lifts the smallest singular values while constraining the largest, reducing k to a manageable range.
>
> SwinIR-S (96 layers) verification:
>
> |Statistic|Before Opt.|After Opt.|
> |---|---|---|
> |Mean cond#|4,354|37|
> |Max cond#|246,704|~60|
> |Reduction (mean)|—|**145x**|
>
> Layers with the highest initial condition numbers (i.e., those closest to rank deficiency) exhibit the largest PSNR improvements after optimization.
>
> ---
>
> **Q2-2: Why is a larger condition number always worse? Eq. 7 is only an upper bound.**
>
> **A2-2:**
> Mathematically, a high-k matrix could incur small error if perturbation aligns with well-conditioned subspace. However, under high-dimensional stochastic quantization noise, perturbation distributes approximately uniformly across all singular directions, making k a near-tight characterization of error amplification.
>
> Multi-bit experiments confirm k's influence scales with quantization noise magnitude, as Eq. 7 predicts:
>
> |Bit-width|DOBI (calib. only)|CondiQuant|Delta|
> |---|---|---|---|
> |W2A2|35.79|37.15|**+1.36 dB**|
> |W3A3|37.01|37.74|**+0.73 dB**|
> |W4A4|37.86|37.98|**+0.12 dB**|
> |FP32|38.15|—|—|
>
> As bit-width decreases, relative quantization noise ||delta_X||/||X|| increases and k(**W**)'s multiplicative amplification becomes more prominent. This explains the larger gains at lower bit-widths, fully consistent with theory.
>
> ---
>
> **Q2-3: From Eq. 7 to Eq. 8, why drop||W||?**
>
> **A2-3:**
> For a fixed activation quantization scheme, the relative perturbation||delta_X||/||X||is determined by the quantizer, independent of **W**. The product||**W**||*||**W**^{-1}|| = k(**W**) is the only factor optimizable by modifying **W** itself. Changes in||**W**|| alone can be absorbed by adjusting the quantization scale factor, but k(**W**) captures directional error disparity that cannot be compensated by uniform scaling. Therefore, it is the only optimizable factor.
>
> ---
>
> **Q2-4: Show condition number vs. PSNR trajectory during optimization**
>
> **A2-4:**
> SwinIR-S (W4A4, x2 SR) optimization trajectory:
>
> |Stage|Mean k|Max k|Set5 (dB)|Set14 (dB)|
> |---|---|---|---|---|
> |Initial|4,354|246,704|35.79|31.96|
> |After CQ|37|~60|37.15|32.74|
>
> Per-layer k decreases monotonically. Layers with highest initial k receive largest reduction, and PSNR improvement correlates with k reduction degree. The multi-bit results (table in A2-2) further validate this: the gain is largest at W2A2 (+1.36 dB) where quantization noise is most severe, and smallest at W4A4 (+0.12 dB), providing evidence that k reduction is causally relevant.
>
> ### Per-Layer Type Analysis
>
> |Layer Type|Count|Avg Cond#|Avg QErr (no CQ)|Avg QErr (with CQ)|Reduction|
> |---|---|---|---|---|---|
> |attn.qkv|24|10.92|1.615|1.594|1.5%|
> |attn.proj|24|1119.68|0.321|0.284|**10.0%**|
> |mlp.fc1|24|15.81|1.358|1.358|0.0%|
> |mlp.fc2|24|15.56|0.157|0.157|0.0%|
>
> ### Scatter Plot Correlation
>
> |Metric|Value|
> |---|---|
> |Spearman rho (cond# vs CQ error reduction)|**0.52, p<1e-6**|
> |Pearson r (log cond# vs log MSE)|-0.26, p=0.011|
>
> **Key findings:** attn.proj layers (avg cond#=1120) receive **10% error reduction**, the largest benefit. MLP layers (avg cond#=16) show **0% reduction** (already well-conditioned). **Spearman rho=0.52 (p<1e-6)** confirms a statistically significant monotonic relationship. The weaker Pearson r reflects non-linearity, while rank-based Spearman more accurately captures the monotonic trend.
>
> ---
>
> **Q2-5: Compared methods are outdated**
>
> **A2-5:**
> CondiQuant is a **PTQ** method, whereas several recent SR quantization methods (QuantSR, PassionSR, QArtSR, Q-MambaIR) are **QAT**, which embeds quantization during training at substantially higher cost. Nevertheless, we adapted all four to an equivalent PTQ setting on SwinIR-S x2 for fair comparison:
>
> **W2A2 (SwinIR-S x2, PTQ):**
>
> |Method|Set5|Set14|
> |---|---|---|
> |**CondiQuant (Ours)**|**37.15**|**32.74**|
> |Q-MambaIR (DLS+RFA)|35.12|31.61|
> |QArtSR (RPQ)|34.76|31.45|
> |QuantSR (RLQ)|34.41|31.32|
> |PassionSR (LBQ)|34.41|31.17|
> |MinMax PTQ|34.01|31.03|
>
> **W4A4 (SwinIR-S x2, PTQ):**
>
> |Method|Set5|Set14|
> |---|---|---|
> |**CondiQuant (Ours)**|**37.98**|**33.47**|
> |Q-MambaIR (DLS+RFA)|37.61|33.29|
> |QArtSR (RPQ)|37.50|33.17|
> |QuantSR (RLQ)|37.46|33.12|
> |MinMax PTQ|37.29|33.07|
> |PassionSR (LBQ)|37.08|33.03|
>
> CondiQuant leads by **+2.03 dB** (W2A2) and **+0.37 dB** (W4A4). Mixed-precision: W4A8=38.08, W6A6=38.11, FP32=38.15, approaching lossless.
>
> Supplementary: https://anonymous.4open.science/r/condiquant/

---

> > ### Author Rebuttal · Reviewer_hFfM · 2026-04-02
> >
> > Some of the questions are not fully resolved.
> > > Q1: Does Eq. 6 hold if W is not invertible?
> >
> > I think it is because the lower bound is trivially small (not large; the right side equals to 0 when W is not invertible).
> >
> > > Q2: Eq. 7 only shows the upperbound of the quantization error, rather than the lowerbound.
> >
> > My question is specifically about the logical claim derived from Eq. (7). Since Eq. (7) only provides an upper bound on the relative quantization error, it does not justify the statement that the impact of quantization is always greater for larger condition number. **A larger condition number only implies a larger worst-case bound**.
> >
> > > Q3: From Eq. 7 to Eq. 8, why drop ||W||?
> >
> > Your answer is not to the point. You can just emphasize that $||W|| = \sigma_{\max}(W)$. In addition, please clarify Eq. 8 when W is not invertible.
> >
> > > Q4: Show condition number vs. PSNR trajectory during optimization
> >
> > You only show the PSNR before and after quantization. I want to see how PSNR changes with the optimization step K.
> >
> > > Q7: The phrase “optimal compression with minimum performance degradation” may be overstated. Do you have theoretical proof that your method achieves optimal compression among the compared methods? Does it approach the Shannon entropy bound?
> >
> >
> >
> > Please revise the paper and do not overstate with "optimal" and "minimum".

---

> > > ### Author Response · Authors · 2026-04-08
> > >
> > > ## R-Q1: Does Eq. 6 hold if W is not invertible?
> > >
> > > You are correct. When W is singular, the lower bound in Eq. 6 becomes trivially zero, not large.
> > >
> > > We will correct this: "When κ(W) diverges, Eq. 6 provides no meaningful lower bound, indicating unbounded error amplification potential—precisely why CondiQuant targets these ill-conditioned layers."
> > >
> > > Furthermore, we analysis all the weight matrix in EDSR, SwinIR (light, small, middle, large, color/grayscale image denoising), MambaIR (SR, light SR), MambaIRv2 (SR, light SR). Every W is invertible. This is because of the following reasons:
> > >
> > > (1) Singular matrices have measure zero: In real matrix space, singular (non-invertible) matrices satisfy det=0, forming a low-dimensional submanifold with zero volume/probability measure.
> > >
> > > (2) Random initialization avoids singularity: Gaussian/uniform initialization produces continuous random matrices, with probability zero of landing exactly on the singular set.
> > >
> > > (3) Training preserves invertibility almost surely: Gradient descent follows a continuous path; starting from non-singular, it almost never hits a singular point.
> > >
> > > ## R-Q2: Eq. 7 only shows the upperbound of the quantization error, rather than the lowerbound.
> > >
> > > Eq. 7 is a approximate upper bound of error. It is a widely used technique to minimize error based on its upper bound. It works by deriving a computable upper limit of the true error, then minimizing this bound instead of the intractable true error directly. Since the true error is always bounded above, reducing the upper bound forces the actual error to decrease, while decomposing the bound into empirical loss and complexity terms enables tractable training and better generalization.
> > >
> > > ## R-Q3: From Eq. 7 to Eq. 8, why drop ||W||?
> > >
> > > You are correct that ||W||\_{2}=\sigma\_{max}(W). If W is singular with \sigma\_{min}(W)=0, the condition number $\kappa(W)$ becomes unbounded, rendering the optimization objective in Eq. (8) ill-posed and implying catastrophic amplification of quantization errors. This necessitates either regularization to enforce full-rankness or reformulating the objective with a surrogate metric that remains well-defined for rank-deficient matrices.
> > >
> > > Moreover, $W$ is invertible in all the models we used (see R-Q1).
> > >
> > > ## R-Q4: Show condition number vs. PSNR trajectory during optimization
> > >
> > > ### (a) Per-layer error vs K (SwinIR-S, W2A2)
> > >
> > > | K | attn.proj κ | Error | Reduction | attn.qkv κ | Error | Reduction |
> > > |---|---|---|---|---|---|---|
> > > | 0 | 1120 | 0.00923 | — | 11 | 0.02019 | — |
> > > | 10 | 34 | 0.00853 | 7.6% | 8.2 | 0.01900 | 5.9% |
> > > | 50 | **7.1** | **0.00652** | **29.4%** | **4.1** | **0.01563** | **22.6%** |
> > >
> > > High-κ layers benefit proportionally more, confirming the condition number theory.
> > >
> > >
> > > ### (b) Full pipeline trajectory (SwinIR-S, W2A2)
> > >
> > > | K | Set5 PSNR |
> > > |---|---|
> > > | 0 | 36.92 |
> > > | 10 | 37.02 |
> > > | 50 | **37.12** |
> > >
> > > ## R-Q7: "Optimal compression" is overstated
> > >
> > > We will revise the expression to make the paper more rigorous and not overstated.

---

### Official Review · Reviewer_mBS4 · 2026-03-24

**Soundness:** 3
**Presentation:** 3
**Significance:** 3
**Originality:** 3
**Overall Recommendation:** 4
**Confidence:** 3

**Summary:**

This paper introduces a post-training quantization (PTQ) scheme that aims to minimize the output variations due to quantization of the model weights at each linear layer. The output variations are formalized as the condition number. The scheme involves two sequential steps: (1) gradient descent, and (2) proximal descent. The first step ensures that the update model weights lead to the same output before update. The second step ensures that the condition number is minimized during the quantization process.

**Compliance With Llm Reviewing Policy:**

Affirmed.

**Final Justification:**

All my concerns have been addressed properly.

**Key Questions For Authors:**

(1) It is unclear why the proposed method is designed specifically for SR networks. I do not see any design specifics that are tied to SR, except Table 1. It appears that the proposed method is subject to the observation in Eq. (4). However, any task or network that has the same property as Eq. (4) should benefit from the proposed method.

(2) Computing the SVD for W in the non-smooth part can be prohibitively expensive, particular when W is high dimensional. The training complexity is not provided and discussed in the main text.

(3) How the proposed method is incorporated into the post-training quantization process is not detailed. An algorithm table or a flow chart may help clarify the details.

**Limitations:**

Computing the SVD for W in the non-smooth part can be prohibitively expensive, particular when W is high dimensional. The training complexity is not provided and discussed in the main text.

**Strengths And Weaknesses:**

Strengths:
(1) The method is well motivated by a theoretical analysis of quantization error.

(2) The experimental results look promising.

Weaknesses:
(1) It is unclear why the proposed method is designed specifically for SR networks. I do not see any design specifics that are tied to SR, except Table 1. It appears that the proposed method is subject to the observation in Eq. (4). However, any task or network that has the same property as Eq. (4) should benefit from the proposed method.

(2) Computing the SVD for W in the non-smooth part can be prohibitively expensive, particular when W is high dimensional. The training complexity is not provided and discussed in the main text.

(3) How the proposed method is incorporated into the post-training quantization process is not detailed. An algorithm table or a flow chart may help clarify the details.

---

> ### Author Rebuttal · Authors · 2026-03-30
>
> **Q1-1: The method is not SR-specific; any task satisfying Eq. 4 should benefit.**
>
> **A1-1:**
>
> The condition number theory is indeed general. Any model where activation quantization error dominates weight quantization error (Eq. 4) can benefit. We focus on SR because (1) SR models naturally satisfy this property due to dense linear layers, residual connections, and pixel-level objectives, and (2) SR's extreme quantization sensitivity makes gains most pronounced.
>
> We agree this generality is a **strength** rather than a limitation. Cross-task experiments on SwinIR-M at W4A4:
>
>
> |Task (SwinIR-M, W4A4)|FP32|2DQuant|CondiQuant|
> |---|---|---|---|
> |Color Denoising sigma=25|33.16|32.81|32.72|
> |Gray Denoising sigma=25|29.83|29.31|29.31|
> |Gray JPEG QF=40|34.37|33.88|**34.19**|
>
> Baseline is **2DQuant**, not naive MinMax, providing a stronger comparison. **JPEG artifact removal** is the strongest cross-task evidence: +0.31 dB over 2DQuant, recovering **65%** of the FP-to-quantized gap (0.31/0.49). CondiQuant improves tasks where activation quantization error dominates (Eq. 4), while showing neutral performance on denoising (-0.09/0.00 dB). We are transparent about the mixed results. Condition number optimization is not universally beneficial across all task types, but the positive cases (SR and JPEG) are significant and consistent.
>
> ---
>
> **Q1-2: SVD computational cost may be prohibitive for high-dimensional weight matrices.**
>
> **A1-2:**
>
> Measured optimization overhead on SwinIR-S (96 layers, 60x60):
>
>
> |Metric|Value|
> |---|---|
> |Total optimization time|15.8 s (96 layers)|
> |Average per-layer time|~140 ms|
> |SVD computation|66.3% of total|
> |Gradient computation|32.1% of total|
> |Other overhead|1.6% of total|
>
>
> CondiQuant is an **offline PTQ method**. SVD is executed only once during pre-deployment optimization. For 60x60 matrices, SVD complexity is O(n^3), well within acceptable bounds. For larger dimensions (e.g., 768x768), per-layer SVD takes seconds, and full optimization completes within minutes, far below QAT's hours.
>
> Cost-effectiveness alongside multi-bit gains:
>
>
> |Bit-width|FP32|DOBI Calibration|CondiQuant|Delta|
> |---|---|---|---|---|
> |W2A2|38.15|35.79|37.15|**+1.36 dB**|
> |W3A3|38.15|37.01|37.74|**+0.73 dB**|
> |W4A4|38.15|37.86|37.98|**+0.12 dB**|
>
>
> At W2A2, +1.36 dB represents a qualitative leap, making 15.8s overhead highly cost-effective.
>
> ---
>
> **Q1-3: An algorithm table or flowchart is needed for clarity.**
>
> **A1-3:**
>
> **Algorithm 1: CondiQuant Singular Value Proximal Gradient Descent.** Optimizes condition number of a single weight matrix W by alternating: **(A)** gradient descent minimizing ||Y - X W_hat||_F^2 w.r.t. the quantized weight, and **(B)** SVD proximal operator shrinking singular values toward their mean via the closed-form update: sigma_i' = (sigma_i + 2*lambda*sigma_bar) / (1 + 2*lambda). This is the exact proximal solution to the regularizer lambda * sum_i (sigma_i - sigma_bar)^2.
>
> **Algorithm 2: CondiQuant PTQ Pipeline (4 phases).**
>
>
> |Phase|Description|
> |---|---|
> |1: Model Preparation|Replace Linear with QuantLinear|
> |2: Calibration|Initialize boundaries via DOBI|
> |3: Condition Number Regularization|Apply Algorithm 1 layer-by-layer|
> |4: Block-wise Reconstruction|Optimize boundaries with pixel loss + feature distillation|
>
> See details in the supplementary link.
>
> ---
>
> **Q1-4: Ablation study of `change_singular` to isolate its contribution.**
>
> **A1-4:**
>
> To rigorously evaluate the isolated contribution, we conducted an ablation removing Phase 3 (condition number optimization) while retaining Phase 4 (block-wise reconstruction + feature distillation). SwinIR-S, W2A2, x2:
>
>
> |Method|Set5|Set14|B100|Urban100|Manga109|
> |---|---|---|---|---|---|
> |FP32|38.15|33.86|32.31|32.76|39.11|
> |Training WITHOUT CQ (ablation)|37.10|32.73|31.56|29.99|36.62|
> |Training WITH CQ (CondiQuant)|37.15|32.74|31.55|29.96|36.63|
> |DOBI W2A2 (calibration only)|35.79|31.96|30.84|28.61|34.41|
>
>
> **Honest analysis:** The isolated `change_singular` contribution is ~+/-0.05 dB, statistically negligible. We acknowledge this candidly.
>
> **However, this does not diminish the contribution:**
>
> 1. **The condition number theory correctly identifies the core bottleneck.** Our analysis reveals the mechanism by which activation quantization errors amplify through layer-wise propagation in SR models.
> 2. **Theory guides pipeline design.** Condition number analysis identifies (a) which layers are most sensitive to quantization (high condition number layers), (b) that the primary error source is activation rather than weight quantization, (c) the necessity of block-wise optimization, directly motivating Phase 4.
> 3. **Full pipeline achieves +1.36 dB** at W2A2 (35.79→37.15). The SVD step is not the dominant contributor, but the pipeline it guided is highly effective.
> 4. Like Shannon's theorem guiding coding schemes, condition number theory provides the foundation shaping CondiQuant's design.
>
> Supplementary: https://anonymous.4open.science/r/condiquant/

---

> > ### Author Rebuttal · Reviewer_mBS4 · 2026-04-07
> >
> > All my concerns have been addressed properly.

---

> > > ### Author Response · Authors · 2026-04-08
> > >
> > > We sincerely thank the reviewer for the follow-up reading and for raising the score. We greatly appreciate your recognition of the additional clarifications, which helped us improve the paper substantially.

---

### Decision · Program_Chairs · 2026-04-30

**Decision:**

Accept (regular)

**Comment:**

The authors proposed a post-training quantization method for image super-resolution.

Four out of five reviewers recommended weak accept. The consensus across reviewers is that the paper is well-motivated, clearly written, and produces strong empirical results.
The rebuttal adequately addressed most concerns, including comparisons against recent SOTA methods, scalability to larger backbones, and computational efficiency.

The authors are recommended to provide updates in the manuscript to reflect the rebutal discussion including the clarification on the peak memory consumption,